# Co-Evolving LLM Coder and Unit Tester via Reinforcement Learning

**Yinjie Wang**[1*], **Ling Yang**[2*†], **Ye Tian**[3], **Ke Shen**[4], **Mengdi Wang**[2]

[1]University of Chicago,   [2]Princeton University,   [3]Peking University,   [4]ByteDance Seed

Project: https://github.com/Gen-Verse/CURE

## Abstract

Mathematical reasoning in large language models has been successfully incentivized through reinforcement learning with verifiable rewards, leading to improved one-shot precision. In this work, we turn our focus to the coding domain. Beyond one-shot precision, we highlight unit test generation as another key factor for enhancing coding ability, since accurate unit tests are essential for enabling self-checking and self-correction during inference. Traditional approaches for fine-tuning LLMs on unit test generation rely heavily on ground-truth code solutions in the training data. We propose CURE, a novel reinforcement learning framework with a dedicated reward design that co-evolves coding and unit test generation capabilities based on their interaction outcomes—without any ground-truth code as supervision. This approach enables flexible and scalable training and allows the unit tester to learn directly from the coder's mistakes. Through extensive evaluations, we demonstrate that our CURE models, derived from base models of varying sizes, excel in both code generation and unit test generation. They naturally extend to downstream tasks such as test-time scaling—achieving a 6.2% improvement over the base model—and agentic unit test generation, with a 25.1% improvement. Our CURE-4B model consistently outperforms Qwen3-4B while achieving 64.8% inference efficiency in unit test generation. Notably, we also find that the CURE model can serve as an effective reward model for reinforcement learning on base models, even in the absence of any labeled supervision.

## 1 Introduction

Recently, the mathematical reasoning capabilities and precision of large language models (LLMs) have seen substantial improvements through post-training optimization techniques such as reinforcement learning [14, 18, 39, 49], as well as through test-time scaling methods guided by reward-based selection strategies [7, 24, 3, 53, 27], including Best of N (BoN). In this paper, we focus on enhancing the coding capabilities of LLMs—a domain critical to the advancement of artificial intelligence—through both post-training optimization and test-time scaling approaches.

Beyond scaling the one-shot coding capabilities of LLMs, we identify generating unit tests as a key factor—and a promising entry point—for improving coding performance. Specifically, we focus on task-derived unit tests, which are generated from a given coding task description and are designed to verify the correctness of the corresponding code. We highlight several advantages of using unit tests in this context. First, their direct alignment with code correctness makes unit tests a reliable reward signal, suitable for guiding both reinforcement learning [52, 8, 22] and test-time scaling or agentic coding pipelines [28, 5, 16, 35, 21]. Second, generated unit tests can be efficiently reused across all candidate solutions during test-time scaling, avoiding the quadratic complexity

---

*Equal Contribution. Contact: yangling0818@163.com

†Corresponding Author

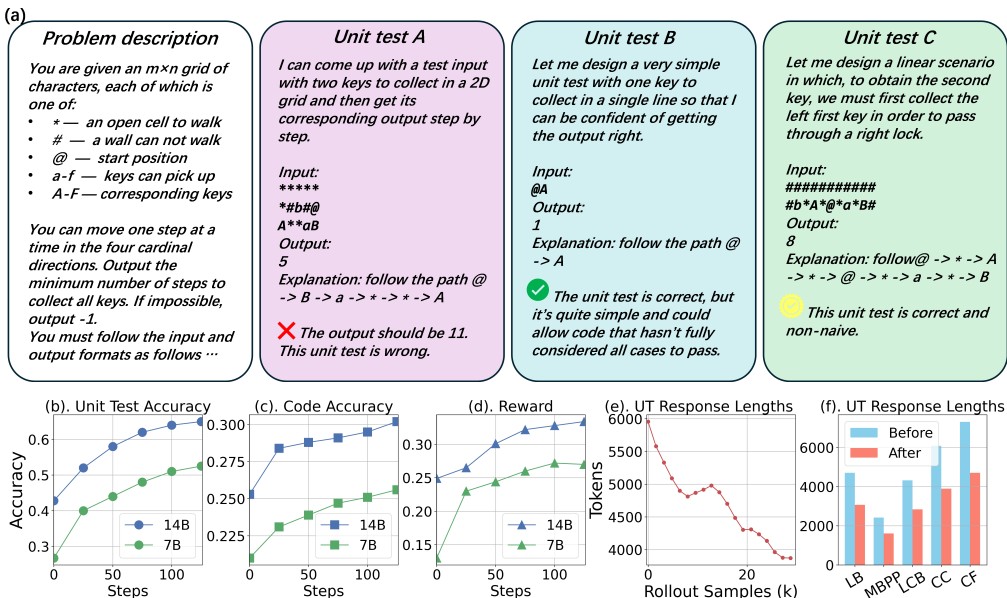

Figure 1: (a). This is an example of a problem description along with three task-derived generated unit tests. The first unit test is incorrect, although it is easily produced due to strong hallucination. The second unit test is correct but naive, allowing some incomplete or unthoughtful code to pass. The final unit test is both correct and non-naive, though generating such a test is much easier than actually solving the full coding problem. (b–d) Co-evolving process: (b) unit test accuracy, (c) code accuracy, and (d) estimated reward versus number of steps. (e-f). The Long CoT unit tester becomes increasingly efficient in reasoning as the response length decreases during optimization.

inherent in scalar or generative reward models, which require separate reward computations for each candidate [7, 24, 3, 53, 27]. Most importantly, generating a unit test does not necessarily require the model to produce a complete solution or algorithm (see Figure 1(a)), substantially simplifying test construction compared to traditional verification approaches, in which LLMs often struggle to verify and correct self-generated solutions [17]. Moreover, using generated unit tests at inference time naturally promotes a self-check and self-correction pattern.

Traditional unit test generation techniques include software analysis methods [11, 30] and machine translation-based approaches [41, 1]. Recent developments show that large language models (LLMs) outperform traditional approaches in unit test generation [38, 50, 36], aided by prompt engineering and agentic techniques [50, 6, 13]. These findings highlight the potential for fine-tuning LLMs to further enhance their unit test generation capabilities [38]. O1-Coder [52] fine-tunes LLMs using unit tests derived from ground-truth code. Inspired by the trade-off between attack rate and accuracy, UTGEN [32] further proposes training LLMs with both correct unit tests from ground-truth code and incorrect tests from perturbed code to enhance downstream inference tasks.

However, training unit test generators in these ways requires supervision from ground-truth code solutions, whose collection is both costly and labor-intensive, thereby limiting the scale and diversity of usable training data. If a unit test generator could instead be trained without reliance on ground-truth code, this would substantially improve the flexibility and scalability of the optimization process. To this end, we propose leveraging the code generator to provide supervision for the unit test generator, while simultaneously improving the code generator itself to produce more accurate outputs that guide the generation of correct unit tests.

Motivated by this, we pose the following central research question for scaling LLMs in coding tasks: ***Can the unit test generator and code generator coevolve effectively, without access to ground-truth code solutions, to improve LLM coding ability?***

We answer this question affirmatively by introducing **CURE**, a novel reinforcement learning framework (Figure 2) that co-evolves a self-play agent acting as both a code generator and a unit test generator. CURE constructs a pairwise reward matrix based on interactions between generated codes and generated tests, enabling mutual supervision and continuous improvement (Figure 1 (b)-(d)). This setup is well-motivated: during reinforcement learning, the coder naturally produces both correct

and incorrect solutions, with the incorrect ones revealing typical failure modes. These, in turn, offer valuable opportunities for the unit test generator to learn to distinguish good code from bad code.

We further demonstrate the utility of the optimized model in two settings. First, and most importantly, it effectively enhances one-shot coding, unit test generation, test-time scaling and agentic coding ability. Second, we find that using the optimized model to generate unit tests, as a reward model for reinforcement learning on the base model, can lead to competitive improvements compared to using ground-truth labeled unit tests. Finally, while long-chain-of-thought (long-CoT) models represent some of the most advanced AI capabilities to date, they suffer from extremely slow inference [48, 42, 14]. To address this, we introduce a response-length-guided transformation on the reward to make the long-CoT unit test generator more efficient in test-time applications.

We summarize our contributions as follows:

1. We propose **CURE**, a novel co-evolving reinforcement learning framework that enables a single model to simultaneously excel at unit test generation and coding, without access to any ground-truth code solutions. The framework employs a theoretically derived and well-motivated reward design for unit test generation. In addition, for long-chain-of-thought models, we introduce a response-length-guided reward transformation to enhance test-time efficiency of the fine-tuned unit test generator. This results in models of different scales: CURE-4B, 7B and 14B.

2. We conduct extensive evaluations on five benchmarks and demonstrate that CURE effectively enhances the abilities of the model in unit test generation and coding, naturally extends to test-time scaling and agentic coding tasks (with a 6.2% average gain in accuracy over the base model), and agentic unit test generation tasks (with a 25.1% average gain in accuracy). Our 4B model consistently outperforms Qwen3-4B while achieving 64.8% inference efficiency in unit test generation.

3. Finally, we show that the trained unit test generator can serve as a reward model to fine-tune LLMs via reinforcement learning—improving coding performance without any human-labeled or ground-truth unit test supervision.

## 2 Related Work

**Unit Test Generation**  Manually creating unit tests is costly and inefficient [5, 25], motivating the development of automated unit test generation methods, such as software analysis methods [11, 30, 9, 12, 33, 10] and traditional neural machine translation approaches [41, 1]. With the recent advancements in LLMs, prompt-based and agentic methods [50, 6, 13] have demonstrated superior performance, further highlighting the potential of training LLMs for unit test generation. In light of this, methods like O1-Coder [52] and UTGEN [32] construct datasets using ground-truth code solutions to fine-tune LLMs for better unit test generation. However, relying on ground-truth code solutions in the training data limits both flexibility and scalability.

**Application of Unit Tests**  Unit tests have been shown to serve as effective rewards for test-time scaling and agentic coding [28]. A common strategy is to generate multiple code and unit test candidates using the model, then select the best-performing sample based on execution results against the generated unit tests [5, 16]. AlphaCodium [35] introduces self-revision by leveraging both public and generated tests to refine solutions. S* [21] further incorporates iterative debugging and pairwise discrimination guided by generated unit tests to enhance final code quality.

**Reinforcement Learning for LLM Improvement**  Proximal Policy Optimization (PPO) [37] uses an actor-critic setup with clipped updates for stability. Direct Preference Optimization (DPO) and its variants [34, 26, 4, 29, 46, 44] skip the critic and directly optimize from preferences using closed-form rewards, improving efficiency. Recent efficient Group Relative Policy Optimization (GRPO) [39] scales well with large-scale reinforcement learning [14, 49, 15, 43]. Reinforcement learning applied specifically to coding tasks has also gained traction [8, 22]. We do not aim to compete with existing reinforcement learning algorithms for code generation; in fact, these RL-on-coding methods can be naturally integrated into our co-evolutionary framework by directly applying them to optimize the coding component.

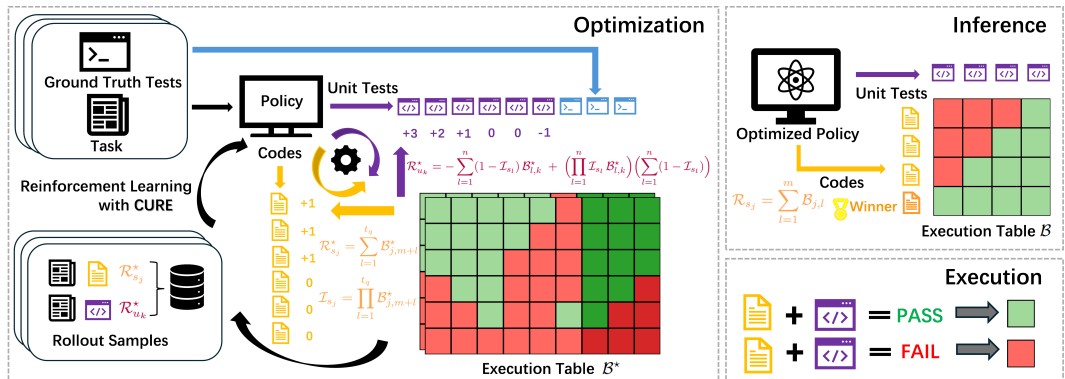

Figure 2: Method Pipeline Overview. In our RL framework, for each task, we generate a batch of unit tests and code solutions, along with some ground-truth unit tests. Using these, we construct an execution table. From this table, we extract rewards for each unit test (Equation 4) and code response (Equation 3). For the long-CoT model, we apply a transformation on the reward to ensure efficiency. Then we optimize both the unit tester and the coder iteratively over time.

## 3 Method

In this section, we begin by formulating our final objective and introducing the general concept of *reward precision* (Section 3.1). We then provide a theoretical analysis of reward precision to derive individual-level rewards for each generated unit test (Section 3.2). Next, we present our novel co-evolving reinforcement learning framework, **CURE**, in Section 3.3. Finally, we introduce a response-length-guided transformation on the reward, designed to improve the efficiency of the unit test generator for long CoT models (Section 3.4).

### 3.1 Motivation: Using Unit Tests for Inference

Unlike mathematical tasks, which are computationally intensive and challenging to verify accurately [17], code-generation tasks benefit significantly from the use of unit tests for efficient verification. It has been shown [28] that the accuracy of code generation can be enhanced by adopting the following BoN approach: For each task $q$, the policy LLM generates $n$ candidate solutions $s_j$, where $1 \leq j \leq n$, and $m$ additional unit tests $u_k$, where $1 \leq k \leq m$. Executing the $n$ generated solutions against these $m$ unit tests produces a binary evaluation matrix $\mathcal{B} \in \{0,1\}^{n \times m}$, where each entry indicates whether a given solution passes a specific test. The reward for solution $s_j$ is defined as follows, and is used to select the optimal coding solution:

$$\mathcal{R}_{s_j} = \sum_{l=1}^{m} \mathcal{B}_{j,l}. \tag{1}$$

Empirically, this reward is typically valid because incorrectly generated unit tests also rarely favor incorrect solutions. However, this assumption can break down when the generated unit tests are of low accuracy, under ambiguous problem formulations, or in binary output tasks. Therefore, we propose our objective for optimizing the unit test generator, **reward precision**:

$$P(\mathcal{R}_{s_{j_1}} > \mathcal{R}_{s_{j_2}} \mid s_{j_1} \text{ is correct, } s_{j_2} \text{ is wrong}). \tag{2}$$

The higher the reward precision, the more accurately the generated unit tests can identify and promote correct solutions. But this is merely an overall objective. To obtain rewards at the individual level for generated unit tests, we conduct the following analysis to derive the reward formulation.

### 3.2 Analysis on Reward Precision

In this section, we identify the key factors that ensure the validity and accuracy of the reward precision defined in Equation 2. Given that the generated responses are i.i.d., we model the binary evaluation results with the following generative process: First, the correctness of a generated solution, denoted by $c_s$, and the correctness of a generated unit test, denoted by $c_u$, are modeled as Bernoulli random variables with success probabilities $p_s$ and $p_u$, respectively. Conditional on their correctness, the execution outcome is another Bernoulli random variable with success probability $p_{c_s c_u}$. Specifically, we have $p_{10} = 0$ and $p_{11} = 1$, while the parameters $p_{00}$ and $p_{01}$ remain unknown.

In the theorem below, we demonstrate increasing the number of generated unit tests $m$ causes the reward precision to converge to 1, provided that certain conditions involving the parameters $p_u$, $p_{00}$, and $p_{01}$ are satisfied. We naturally derive our optimization objective with this theoretical analysis.

**Theorem 3.1.** *Consider a ground truth unit test $u_k$, a correct solution $s_{j_1}$, and an incorrect solution $s_{j_2}$. The precision based on a single ground truth test is given by $P(\mathcal{B}_{j_1,k} > \mathcal{B}_{j_2,k}) = 1 - P(\text{the incorrect solution } s_{j_2} \text{ passes test } u_k)$. However, when using the aggregated reward defined in Equation 1, we have $P(\mathcal{R}_{s_{j_1}} > \mathcal{R}_{s_{j_2}}) \to 1$ as $m \to \infty$, if and only if $\mu > 0$, where*

$$\mu := p_u(1 - p_{01}) - (1 - p_u)p_{00}.$$

*Moreover, under this condition, the reward precision satisfies*

$$P(\mathcal{R}_{s_{j_1}} > \mathcal{R}_{s_{j_2}}) \gtrsim 1 - e^{-\mu^2 m/8}.$$

From this theorem, we observe that $\mu$ not only guarantees the convergence and validity of the aggregated reward (Equation 2), but also governs the rate at which it converges to 1. Specifically, a larger value of $\mu$ implies that fewer unit tests are needed to obtain a reliable reward signal.

Therefore, we use $\mu$ as the optimization objective for the unit test generator, estimating the individual value of $\mu$ for each unit test from the execution matrix to serve as its reward. Intuitively, optimizing $\mu$ corresponds to increasing the accuracy $p_u$ while controlling the error rates $p_{01}$ and $p_{00}$ for the generated unit tests. We now introduce our algorithm to co-evolve the coder and the unit tester.

### 3.3 Co-evolving Coder and Unit Tester with RL

For each task $q$ in the training set, which is paired with $t_q$ ground truth unit tests, the policy LLM generates $n$ candidate solutions and $m$ additional unit tests $u_k$, where $1 \le k \le m$. Similarly, we obtain a binary evaluation matrix $\mathcal{B}^\star \in \{0,1\}^{n \times (m+t_q)}$ by executing the $n$ generated solutions against these $m + t_q$ unit tests. The last $t_q$ columns correspond to the ground truth unit tests. This evaluation matrix serves as the basis for estimating rewards for both the solution generator and the unit test generator, enabling joint optimization via reinforcement learning.

For solution $s_j$, where $1 \le j \le n$, we assign higher rewards to solutions that pass more ground-truth unit tests, reflecting greater correctness and generalizability. The reward is defined as:

$$\mathcal{R}_{s_j}^\star = \sum_{l=1}^{t_q} \mathcal{B}_{j,m+l}^\star. \tag{3}$$

For each generated unit test $u_k$, where $1 \le k \le m$, we estimate the reward $\mu = p_u(1 - p_{01}) - (1 - p_u)p_{00}$ from the execution matrix $\mathcal{B}^\star$ by deriving estimators for $p_u$, $p_{01}$, and $p_{00}$. This leads to the following form of the estimated individual-level reward:

$$\mathcal{R}_{u_k}^\star = -\sum_{l=1}^{n} (1 - \mathcal{I}_{s_l}) \mathcal{B}_{l,k}^\star + \left( \prod_{l:\mathcal{I}_{s_l}=1} \mathcal{B}_{l,k}^\star \right) \left( \sum_{l=1}^{n} (1 - \mathcal{I}_{s_l}) \right), \tag{4}$$

where $\mathcal{I}_{s_j} = \prod_{l=1}^{t_q} \mathcal{B}_{j,m+l}^\star$. The detailed derivation is provided in Appendix A. Intuitively, $\mathcal{R}_{u_k}^\star$ is positive and proportional to the number of incorrect solutions that fail test $u_k$ when $u_k$ correctly passes all accurate solutions. Conversely, $\mathcal{R}_{u_k}^\star$ is negative and proportional to the number of incorrect solutions that pass test $u_k$ when $u_k$ fails even one correct solution. Here, a correct solution is defined as one passing all ground-truth unit tests, whereas an incorrect solution fails at least one ground-truth test. Therefore, this theoretically derived reward serves as an effective objective, optimizing the accuracy and discriminative power of generated unit tests. Naively using reward functions like "whether the unit test passes all correct codes" incentivize the generation of trivial or overly permissive tests that simply maximize pass rates. This undermines the reliability of the reward signal and diminishes the overall effectiveness of the co-evolution process.

After collecting the rollout samples for codes and unit tests and their rewards, we optimize the policy with the following objective iteratively:

$$\mathcal{J}(\theta, \{o_i\}_{i=1}^G) = \mathbb{E}_{\substack{q \sim P(Q) \\ o_i \sim \pi_{\theta_{\text{old}}}(\cdot|q)}} \left[ \frac{1}{G} \sum_{i=1}^{G} \sum_{t=1}^{|o_i|} C_\epsilon \big( \frac{\pi_\theta(o_{i,t} \mid q, o_{i,<t})}{\pi_{\theta_{\text{old}}}(o_{i,t} \mid q, o_{i,<t})}, A_{o_i} \big) \right]$$
$$- \mathbb{E}_{\substack{q \sim P(Q) \\ o_i \sim \pi_{\theta_{\text{old}}}(\cdot|q)}} \left[ \beta \, \mathrm{D}_{\text{KL}} \big[ \pi_\theta \, \| \, \pi_{\text{ref}} \big] \right],$$

where $C_\epsilon(r, A) := \min(rA, \text{clip}(r, \varepsilon)A)$, $\text{clip}(r, \varepsilon) := \min(\max(r, 1 - \varepsilon), 1 + \varepsilon)$, $\pi_\theta$ is the policy to be optimized, $\pi_{\text{old}}$ is the old policy, $\{o_i\}_{i=1}^G$ are the rollout responses, and $A_{o_i}$ is the normalized reward corresponding to $\mathcal{R}_{o_i}^\star$. Specifically, we iteratively optimize the policy for coding ability with $\mathcal{J}(\theta, \{s_j\}_{j=1}^n)$, and unit test generation ability with $\mathcal{J}(\theta, \{u_k\}_{k=1}^m)$ (see Figure 2).

## 3.4 Improve Efficiency of Long-CoT Unit Tester

In addition to experiments conducted on base LLMs, we also perform experiments using the long-CoT model, which currently exemplifies the highest reasoning capabilities of LLMs. However, it is well-documented that these long-CoT models suffer from significantly increased inference times [48, 42, 14]. To enhance efficiency, we propose a general response-length-aware transformation applied to the rewards of unit tests specifically when utilizing long-CoT models.

Formally, for each task $q$, consider a set of standardized rewards $\{r_i\}_{i=1}^m$ (standardized by subtracting the mean) and the corresponding response lengths $\{l_i\}_{i=1}^m$. Our goal is to assign negative values to overly long responses proportionally to their lengths while ensuring that the transformed rewards maintain a clear separation such that negative original rewards remain negative and positive original rewards remain positive. Specifically, we first transform the rewards to $\widehat{r}_i$ by

$$\widehat{r}_i = \begin{cases} -l_i + T_l & \text{if } r_i > 0, \\ -l_{\max} + T_l & \text{if } r_i \leq 0, \end{cases}$$

where $T_l = \text{median}\{l_j \mid r_j > 0\}$, $l_{max} = \max\{l_j \mid r_j > 0\}$. Subsequently, we balance the transformed rewards between positive and negative responses and normalize them, yielding the final transformed reward $r_i^\star$, defined by $r_i^\star = \alpha \widehat{r}_i / \sigma$ if $\widehat{r}_i > 0$, or $r_i^\star = \widehat{r}_i / \sigma$ if $\widehat{r}_i \leq 0$, where $\alpha = \sum_{j:\widehat{r}_j<0}(-\widehat{r}_j)/(\sum_{j:\widehat{r}_j>0} \widehat{r}_j)$, and $\sigma$ is the standard deviation calculated over the set $\{\alpha \widehat{r}_i \mid \widehat{r}_i > 0\} \cup \{\widehat{r}_i \mid \widehat{r}_i \leq 0\}$. In this way, we aim to preserve the original reward information to some extent, while penalizing overly long responses.

# 4 Experiments

## 4.1 Settings

**Datasets** We select five widely used coding datasets for our comprehensive evaluation: LiveBench [45], MBPP [2], LiveCodeBench [19], CodeContests [23], and CodeForces [31]. Specifically, for CodeContests, we extract tasks with difficulty level $\leq 2$, and randomly split them into a training set of 4.5k examples and an evaluation set of 200 examples. For LiveCodeBench, we utilize version 2, which contains 511 problems. For MBPP, we use its standard test set for evaluation. The CodeForces data used in our experiments has no overlap with CodeContests [31]; we randomly sample 500 examples from it for evaluation.

**Models and Optimization** We use Qwen2.5-7B and 14B [47] as our standard base models, and select Qwen3-4B as the base model for the long-CoT variant. At each sampling step during reinforcement learning, we generate 32 rollouts for unit tests and 32 for code using vLLM [20], with a temperature of 1.0, top-$p$ of 0.95, and top-$k$ of 40. For optimization, we set the learning rate to $1 \times 10^{-6}$ and the KL coefficient to 0.01. Specifically, for the long-CoT model, we use a lower temperature of 0.8 and apply a response-length-guided transformation to the unit test reward to improve post-training inference efficiency. We train these models using 8 A100 GPUs.

Table 1: Performance of CURE models and baseline models across five benchmarks. Each entry reports the average accuracy (%) of generated unit tests (UT), the average one-shot code generation accuracy (Code), and the Best-of-N (BoN) accuracy, using 16 generated code solutions and 16 generated unit tests. "Long" refers to the long-CoT models. The Coder models here are also instruction-finetuned models.

| Model | LiveBench | | | MBPP | | | LiveCodeBench | | | CodeContests | | | CodeForces | | |
|---|---|---|---|---|---|---|---|---|---|---|---|---|---|---|---|
| | UT | Code | BoN | UT | Code | BoN | UT | Code | BoN | UT | Code | BoN | UT | Code | BoN |
| Qwen2.5-14B-Coder | 39.0 | 42.2 | 53.1 | 75.1 | 72.6 | 84.9 | 41.6 | 38.2 | 47.7 | 37.3 | 23.3 | 32.0 | 22.1 | 7.8 | 13.5 |
| Qwen2.5-14B-Ins | 27.8 | 36.4 | 51.7 | 72.8 | 76.3 | 83.2 | 35.7 | 33.5 | 45.1 | 43.8 | 25.6 | 33.4 | 20.7 | 7.3 | 12.5 |
| **CURE-14B** | **55.4** | **45.2** | **57.0** | **85.3** | **78.1** | **85.4** | **63.1** | **39.9** | **48.7** | **64.4** | **30.2** | **38.9** | **53.5** | **10.2** | **19.1** |
| Qwen2.5-7B-Coder | 19.3 | 35.0 | 42.9 | 41.3 | 68.0 | 79.6 | 20.6 | 29.8 | 34.8 | 12.9 | 22.8 | 23.8 | 7.2 | 6.7 | 9.1 |
| Qwen2.5-7B-Ins | 26.5 | 31.1 | 35.9 | 35.8 | 66.3 | 79.4 | 28.6 | 26.9 | 32.6 | 26.7 | 21.2 | 25.8 | 18.9 | 5.4 | 8.9 |
| **CURE-7B** | **44.2** | **37.3** | **45.5** | **74.5** | **69.4** | **82.3** | **48.7** | **31.6** | **41.8** | **52.5** | **25.7** | **29.7** | **39.4** | **7.7** | **10.8** |
| Qwen3-4B (Long) | 36.8 | 72.5 | 78.1 | 76.5 | 88.4 | 90.1 | 50.9 | 74.5 | 80.0 | 43.6 | 53.1 | 58.1 | 54.1 | 28.8 | 38.5 |
| **CURE-4B** (Long) | **84.6** | **74.6** | **82.0** | **83.3** | **89.5** | **91.2** | **86.8** | **75.9** | **80.6** | **72.2** | **55.4** | **59.9** | **65.8** | **31.3** | **40.2** |

Table 2: Application to GPT-series models. We apply CURE-4B as a unit tester to scale GPT models serving as coders, achieving improved performance while maintaining cost efficiency. The two entries report the average API cost (Cost, in units of $10^{-3}$ USD) per task and the overall accuracy (Acc) for each benchmark.

| Model | LB | | MBPP | | LCB | | CC | | CF | |
|---|---|---|---|---|---|---|---|---|---|---|
| | Cost | Acc | Cost | Acc | Cost | Acc | Cost | Acc | Cost | Acc |
| 4o (one-shot) | 4.8 | 48.4 | 2.7 | 85.0 | 5.8 | 48.7 | 5.5 | 41.0 | 7.1 | 11.1 |
| 4o-mini (one-shot) | 0.3 | 46.3 | 0.2 | 80.1 | 0.4 | 44.3 | 0.3 | 38.8 | 0.4 | 12.0 |
| 4o-mini (BoN-16) | 10.8 | 55.4 | 6.7 | 81.5 | 12.0 | 50.7 | 10.1 | 40.6 | 13.1 | 13.5 |
| 4o-mini-**CURE**(BoN-16) | 4.7 | **58.6** | 2.7 | **86.1** | 5.6 | **56.8** | 5.3 | **46.4** | 6.9 | **21.2** |
| 4.1-mini (one-shot) | 0.6 | 65.4 | 0.3 | 88.4 | 0.6 | 68.1 | 1.0 | 51.3 | 1.5 | 22.8 |
| 4.1-mini (BoN-16) | 32.5 | 69.5 | 14.7 | 88.2 | 31.9 | 73.4 | 42.2 | 56.9 | 59.5 | 34.1 |
| 4.1-mini-**CURE** (BoN-16) | 9.3 | **74.2** | 4.6 | **89.6** | 9.6 | **74.1** | 15.5 | **58.1** | 24.4 | **35.1** |

**Test-time Scaling and Agentic Coding**    Best-of-N (BoN) is the most straightforward and widely used test-time scaling and agentic coding method [28, 5], and serves as a primary metric for evaluating coding performance in our setting. Specifically, the policy generates $n$ candidate code solutions and $m$ unit tests, then selects the best solution based on the reward defined in Equation 1. We also evaluate our approach under several other agentic coding and test-time scaling pipelines [16, 35, 21]. In particular, MPSC [16] generates multiple code solutions, unit tests, and specifications per task, and selects the best solution by computing a consistency score. AlphaCodium [35] generates comprehensive unit tests to critique the generated solutions and iteratively refine the code accordingly. S* [21] organically combines iterative debugging using public unit tests and generates unit tests for pairwise discrimination, in order to select the most promising solution. See details in Appendix C.4.

**Agentic Unit Test Generation**    We also evaluate our model's utility in an agentic unit test generation pipeline. Following prior work [50, 6], we first generate unit tests and then iteratively refine them based on their execution results on the corresponding code. See details in Appendix C.5.

## 4.2 Results

**CURE significantly improves the overall coding ability.**    Specifically, we apply our optimization to derive the CURE-7B and CURE-14B models from the base Qwen2.5-7B-Instruct and Qwen2.5-14B-Instruct models. Figure 1 (b–d) show the co-evolution process for unit test accuracy, code accuracy, and estimated reward, demonstrating a stable and promising co-evolving pattern. The resulting CURE models surpass their respective base models on average by 24.4% in unit test accuracy, 4.5% in one-shot code generation accuracy, and 5.1% in Best-of-N (BoN) accuracy (using 16 code solutions and 16 unit tests) (Table 1). Notably, CURE also consistently outperforms the corresponding coding-supervised fine-tuned (SFT) models—Qwen2.5-Coder-Instruct—across all three metrics. Moreover, our results show that the optimization leads to consistent and robust improvements across various BoN settings (Figure 3 (a)). This indicates that the CURE models not only enhance the overall performance ceiling (when large amounts of code and unit test samples are

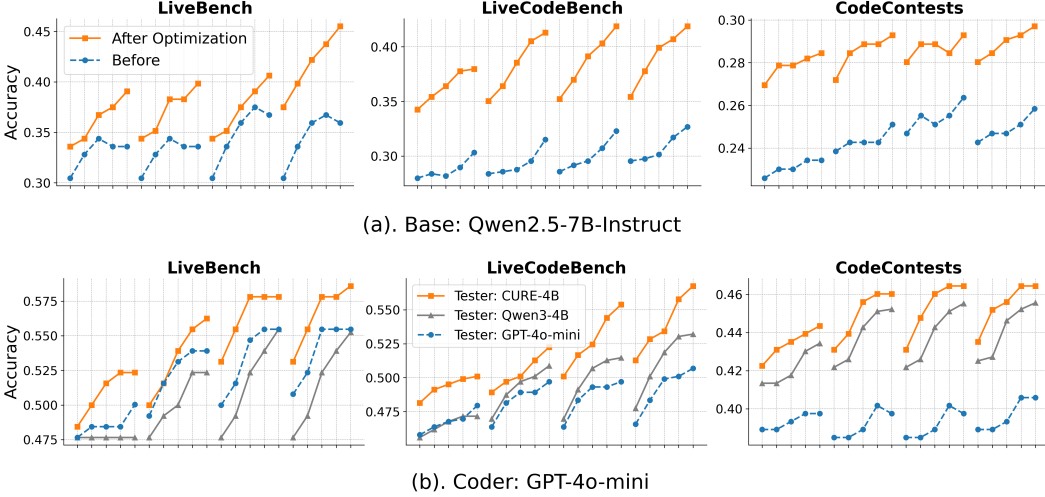

(a). Base: Qwen2.5-7B-Instruct

(b). Coder: GPT-4o-mini

Figure 3: The BoN performance improvement across benchmarks. Four curves (left to right) show sampling 2, 4, 8, and 16 generated codes; each curve's five points represent 1, 2, 4, 8, and 16 generated unit tests. (a). Improvement in BoN performance on open-source models after optimization. The model serves as both coder and unit tester here. (b). BoN improvement with optimized unit tester on GPT-series coders.

generated), but also improve self-check efficiency in low-sample regimes (e.g., when sampling only 1 or 2 candidates).

**Robust for long-CoT models and achieve inference efficiency.** We also evaluate CURE's optimization on the Long-CoT model, Qwen3-4B, incorporating our response-length-guided reward transformation. The resulting CURE-4B model consistently outperforms Qwen3-4B in unit test accuracy, code accuracy, and BoN accuracy (Table 1). Notably, the average response length for unit test generation is reduced to 64.8% of its original length (Figure 1 (e-f)), significantly improving inference-time efficiency. We also observe that the accuracy gains for standard base models are more substantial than for long-CoT models, which aligns with the demonstrated findings [51]. Long-CoT models have already captured much of the benefit from scaling through CoT reasoning and gain less from BoN compared to standard models.

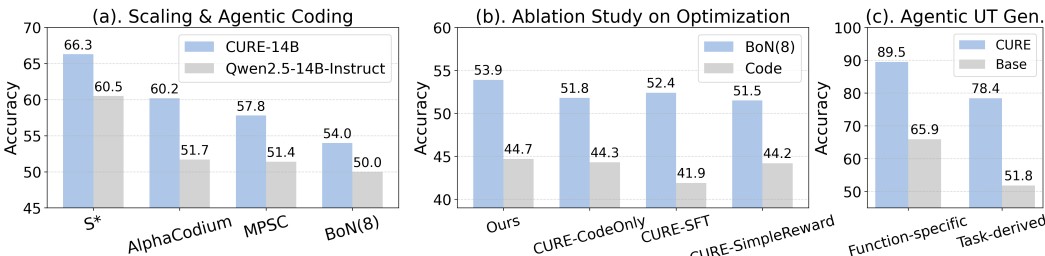

Figure 4: (a). Application of CURE to various test-time scaling and agentic coding methods. We set the number of generated samples to eight in the BoN setting here. (b). Ablation study on optimization strategies and reward design choices, using Qwen2.5-14B-Instruct as the base model. All training runs are conducted with 100 optimization steps. (c). Application of CURE to different agentic unit test generation tasks. "Function-specific" refers to tasks where the input includes both the problem description and the ground-truth code, whereas "Task-derived" refers to tasks where the input consists solely of the problem description. (a–c) are all evaluated on LiveBench, with Qwen2.5-14B-Instruct used as the base model.

**CURE models help API-inference models become more powerful and cost-efficient.** We apply CURE-4B as the unit tester and evaluate its effect when paired with GPT-series models as coders, to disentangle the effects of the long-CoT coders' strong coding ability from the unit test generation ability. We find that CURE improves the BoN accuracy of GPT-4o-mini and GPT-4.1-mini by an average of 5.5% and 1.8%, respectively (Table 2). Notably, using GPT-4o-mini as the coder and

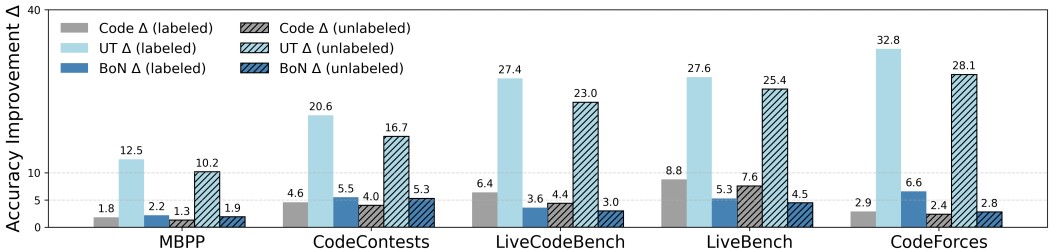

Figure 5: Accuracy improvement of Qwen2.5-14B-Instruct when trained with reinforcement learning using labeled unit tests as rewards versus using CURE-generated unit tests as rewards. Both models are trained for 150 steps. The BoN setting involves generating 16 samples for both code and unit tests.

CURE-4B as the unit tester yields a 7.0% improvement over GPT-4o one-shot performance, while also reducing cost. This demonstrates our model's strong potential for reducing the cost of API-based pipelines. In contrast, scaling GPT-4o-mini alone results in only a 1.5% gain while incurring nearly twice the API cost compared to using CURE-4B. As shown in Figure 3(b), CURE-4B consistently outperforms both Qwen3-4B and GPT-4o-mini as a unit tester across different BoN settings. These results demonstrate the effectiveness of using unit tests generated by the CURE model.

**Serves as an effective reward model enabling RL without any labeled data.** We have already demonstrated the utility of unit tests generated by the CURE model for solution selection. But can the CURE model also serve as a reward model to guide reinforcement learning? We apply CURE-4B to generate unit tests as supervision for reinforcement learning training on the Qwen2.5-14B-Instruct model. Surprisingly, the resulting performance improvements are comparable to those achieved using ground-truth labeled supervision, across all three metrics: code generation accuracy, unit test accuracy, and BoN accuracy (Figure 5). This demonstrates that CURE can serve as an effective reward model not only for inference-time enhancement but also for guiding optimization during training.

**Broad application to test-time scaling and agentic coding methods.** In addition to the standard test-time scaling method BoN [28, 5], we also evaluate CURE-14B on several other test-time scaling and agentic methods—MPSC [16], AlphaCodium [35], and S* [21]—achieving an average improvement of 6.2% over the base model Qwen2.5-14B-Instruct (Figure 4(a)). Beyond code and unit test generation, these pipelines involve iterative refinement and debugging based on execution results, which require comprehensive coding and self-correction capabilities—capabilities our model successfully demonstrates. We further evaluate CURE on agentic unit test generation tasks, which focus on refining unit tests based on execution results from code, and observe an average improvement of 25.1% in unit test accuracy over the base model (Figure 4(c)).

**Ablation study on optimization methods and reward designs.** We conduct ablation studies on two aspects of the optimization process. First, we conduct experiments optimizing only the coder and using supervised fine-tuning (selecting the samples with positive rewards to fine-tune) instead of reinforcement learning. Second, we evaluate a simplified reward design for the unit test: assigning a reward of 1 if all correct codes pass, and 0 otherwise, which is an estimate of $p_u$. We find that CURE consistently outperforms these alternatives and remains the optimal choice across all ablations (Figure 4(b)). Optimizing only for code generation does not improve the model's ability to produce accurate unit tests and therefore falls short in self-check-based inference scaling (e.g., BoN). Supervised fine-tuning focuses solely on positive examples, ignoring informative negative samples. Moreover, using a simple reward during optimization leads to poor control over key error probabilities: $p_{01}$ and $p_{00}$ reach 40.5% and 15.8%, respectively. In contrast, our theoretically derived reward better constrains these values to 30.1% and 10.6%, improving the precision of selection and the overall effectiveness of solution ranking.

## 5    Discussions

In this paper, we propose CURE, a novel optimization framework combined with a theoretically derived reward for the unit tester, that co-evolves models' coding and unit test generation capabilities without requiring any ground-truth code for supervision, which greatly enhances flexibility and scala-

bility. Through extensive evaluations on five benchmarks, our results demonstrate that CURE models achieve significant performance improvements in both code generation and unit test generation tasks. Our long-CoT model CURE-4B consistently outperforms Qwen-4B while achieving significantly higher efficiency in unit test generation. Moreover, CURE proves effective in broader applications, including test-time scaling and agentic coding (6.2% improvement), agentic unit test generation (25.1% improvement), and as a reward model for reinforcement learning.

CURE currently focuses on Python competition-style tasks evaluated via stdin/stdout unit tests. Extending CURE to support functional tests and additional programming languages is essential for greater practical utility. In addition, CURE still depends on ground-truth unit tests during reinforcement learning to achieve high performance. Removing this reliance while preserving comparable performance remains an intriguing direction for future work.

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

# A Proofs of Theoretical Results

*Proof.* (of Theorem 3.1)

**Set–up and intuition.** For every test index $k(1 \leq k \leq m)$ define

$$X_k := \underbrace{\mathcal{B}_{j_1 k}}_{\text{outcome on correct } s_{j_1}} - \underbrace{\mathcal{B}_{j_2 k}}_{\text{outcome on wrong } s_{j_2}} \in \{-1, 0, 1\}.$$

Positive $X_k$ means the correct solution beats the wrong one on test $k$, $X_k = 0$ means they tie, and $X_k = -1$ means the wrong solution wins. The reward difference after $m$ tests is $D_m := \sum_{k=1}^{m} X_k = \mathcal{R}_{s_{j_1}} - \mathcal{R}_{s_{j_2}}$. Our target event $\{\mathcal{R}_{s_{j_1}} > \mathcal{R}_{s_{j_2}}\}$ coincides with $\{D_m > 0\}$, so we analyse the sign of $D_m$.

**Single ground-truth test.** Assume a particular test $u_k$ is *correct* (i.e. $c_{u_k} = 1$). Because a correct solution *always* passes a correct test ($p_{11} = 1$) we have $\mathcal{B}_{j_1 k} = 1$ with probability 1. Conversely, an incorrect solution passes that same correct test with probability $p_{01}$, so

$$P[\mathcal{B}_{j_2 k} = 0] = 1 - p_{01}.$$

Hence

$$P[X_k = 1] = P[\mathcal{B}_{j_1 k} = 1, \ \mathcal{B}_{j_2 k} = 0] = 1 - p_{01}, \quad P[X_k \leq 0] = p_{01}.$$

Therefore $P(X_k > 0) = 1 - p_{01}$, which proves the first statement.

**Distribution of $X_k$.** Let $I_k := \mathbf{1}\{c_{u_k} = 1\}$ indicate whether the $k$-th test is correct. By the data-generation assumption,

$$P(I_k = 1) = p_u, \qquad P(I_k = 0) = 1 - p_u.$$

*Case $I_k = 1$:* we are in the setting of Step 1, so

$$P(X_k = 1 \,|\, I_k = 1) = 1 - p_{01}, \quad P(X_k = -1 \,|\, I_k = 1) = 0, \quad P(X_k = 0 \,|\, I_k = 1) = p_{01}.$$

*Case $I_k = 0$:* the test itself is wrong. Now a correct solution fails with probability 1 ($p_{10} = 0$), while the incorrect solution can pass spuriously with probability $p_{00}$. Thus

$$P(X_k = 1 \,|\, I_k = 0) = 0, \ P(X_k = -1 \,|\, I_k = 0) = p_{00}, \ P(X_k = 0 \,|\, I_k = 0) = 1 - p_{00}.$$

Applying the law of total probability yields the unconditional mass

$$P(X_k = 1) = p_u(1 - p_{01}) + (1 - p_u) \cdot 0 = p_u(1 - p_{01}),$$

$$P(X_k = -1) = (1 - p_u)p_{00},$$

$$P(X_k = 0) = 1 - P(X_k = \pm 1).$$

Denote

$$\mu := E[X_k] = 1 \cdot P(X_k = 1) + (-1) \cdot P(X_k = -1) = p_u(1 - p_{01}) - (1 - p_u)p_{00},$$

$$\sigma_k^2 := \text{Var}(X_k) = E[X_k^2] - \mu^2 = P(X_k = 1) + P(X_k = -1) - \mu^2.$$

All $X_k$'s are i.i.d. because the unit tests are generated independently and the solutions themselves are fixed.

**Convergence Analysis.** Write the empirical mean $\overline{X}_m := \frac{1}{m} \sum_{k=1}^{m} X_k$. Since $E[X_k] = \mu$ and $E[|X_k|] \leq 1$, the strong law of large numbers (SLLN) tells us

$$\overline{X}_m \xrightarrow{\text{a.s.}} \mu \quad (m \to \infty).$$

But $D_m/m = \overline{X}_m$, hence

$$\frac{D_m}{m} \xrightarrow{\text{a.s.}} \mu.$$

*Consequences.*

- If $\mu > 0$, then $\frac{D_m}{m}$ is eventually positive almost surely, so $P(D_m > 0) \to 1$.

- If $\mu < 0$, $\frac{D_m}{m}$ is eventually negative a.s., so $P(D_m > 0) \to 0$.

- If $\mu = 0$, $\frac{D_m}{\sqrt{m}}$ has variance $\sigma_k^2$ and remains $O_p(1)$, whence $P(D_m > 0) \to \frac{1}{2}$ by symmetry of the CLT limit distribution.

**Explicit tail bound for finite $m$, assuming $\mu > 0$.**

Recall $X_k \in \{-1, 0, 1\}$ and $E[X_k] = \mu > 0$. Define the centred variables

$$Z_k := X_k - \mu \quad (1 \le k \le m),$$

so that $E[Z_k] = 0$. Because $-1 \le X_k \le 1$, we have $-1 - \mu \le Z_k \le 1 - \mu$. Since $\mu \in (0, 1)$, we have

$$|Z_k| \le 2 \quad \text{almost surely.}$$

Now we apply Hoeffding's additive inequality. Let $Z_1, \ldots, Z_m$ be independent, centred random variables satisfying $|Z_k| \le c$ a.s. for every $k$. For any $t > 0$,

$$P\Big(\sum_{k=1}^m Z_k \le -t\Big) \le \exp\Big(-\frac{t^2}{2mc^2}\Big). \tag{Hoeffding}$$

Here $c = 2$, By definition

$$D_m = \sum_{k=1}^m X_k = \sum_{k=1}^m (Z_k + \mu) = m\mu + \sum_{k=1}^m Z_k.$$

Hence

$$\{D_m \le 0\} = \Big\{\sum_{k=1}^m Z_k \le -m\mu\Big\}.$$

Substituting $t = m\mu$ and $c = 2$ into (Hoeffding) gives

$$P(D_m \le 0) = P\Big(\sum_{k=1}^m Z_k \le -m\mu\Big) \le \exp\Big(-\frac{(m\mu)^2}{8m}\Big) = \exp\Big(-\frac{\mu^2 m}{8}\Big).$$

Finally,

$$P(D_m > 0) = 1 - P(D_m \le 0) \ge 1 - \exp\Big(-\frac{\mu^2 m}{8}\Big),$$

yielding the advertised exponential guarantee.

$\square$

**Proposition A.1.** *Given the execution table, the individual reward for unit test $u_k$ can be estimated by*

$$\mathcal{R}_{u_k}^\star = -\sum_{l=1}^n (1 - \mathcal{I}_{s_l})\mathcal{B}_{l,k}^\star + \Big(\prod_{l:\mathcal{I}_{s_l}=1} \mathcal{B}_{l,k}^\star\Big)\Big(\sum_{l=1}^n (1 - \mathcal{I}_{s_l})\Big).$$

*Proof.* (of Proposition A.1) We use the following estimation to detect if a code solution $s_j$ is correct or not:

$$\mathcal{I}_{s_j} = \prod_{l=1}^{t_q} \mathcal{B}_{j,m+l}^\star.$$

So the accuracy of $u_k$, $\widehat{p}_u$, can be estimated by

$$\prod_{l:\mathcal{I}_{s_l}=1} \mathcal{B}_{l,k}^\star.$$

Similarly, we can obtain estimator $1 - \widehat{p}_{01}$ and $\widehat{p}_{00}$ as

$$\sum_{l=1}^n (1 - \mathcal{I}_{s_l})(1 - \mathcal{B}_{l,k}^\star)/\sum_{l=1}^n (1 - \mathcal{I}_{s_l}), \quad \sum_{l=1}^n (1 - \mathcal{I}_{s_l})\mathcal{B}_{l,k}^\star/\sum_{l=1}^n (1 - \mathcal{I}_{s_l}),$$

respectively. Finally, we derive $\widehat{\mu} = \widehat{p_u}(1 - \widehat{p}_{01}) - (1 - \widehat{p_u})\widehat{p}_{00}$:

$$\left[ (\prod_{l:\mathcal{I}_{s_l}=1} \mathcal{B}_{l,k}^\star)(\sum_{l=1}^{n}(1 - \mathcal{I}_{s_l})(1 - \mathcal{B}_{l,k}^\star)) - (1 - \prod_{l:\mathcal{I}_{s_l}=1} \mathcal{B}_{l,k}^\star)(\sum_{l=1}^{n}(1 - \mathcal{I}_{s_l})\mathcal{B}_{l,k}^\star) \right] / \sum_{l=1}^{n}(1 - \mathcal{I}_{s_l})$$

$$= \left[ -\sum_{l=1}^{n}(1 - \mathcal{I}_{s_l})\mathcal{B}_{l,k}^\star + (\prod_{l:\mathcal{I}_{s_l}=1} \mathcal{B}_{l,k}^\star)(\sum_{l=1}^{n}(1 - \mathcal{I}_{s_l})) \right] / \sum_{l=1}^{n}(1 - \mathcal{I}_{s_l}).$$

Given that $\sum_{l=1}^{n}(1 - \mathcal{I}_{s_l})$ is constant for different $k$, we have our final reward for $u_k$:

$$-\sum_{l=1}^{n}(1 - \mathcal{I}_{s_l})\mathcal{B}_{l,k}^\star + (\prod_{l:\mathcal{I}_{s_l}=1} \mathcal{B}_{l,k}^\star)(\sum_{l=1}^{n}(1 - \mathcal{I}_{s_l})).$$

$\square$

# B  Additional Experimental Results

Table 3: This is the error analysis table corresponding to Table 1. Each cell reports the "accuracy improvement over the base model (standard error)." Note that the accuracies of unit test and code are evaluated over 16 independent runs, whereas BoN scaling is computationally intensive, so we report BoN accuracy based on a single run per benchmark.

| Model | LiveBench | | MBPP | | LiveCodeBench | | CodeContests | | CodeForces | |
|---|---|---|---|---|---|---|---|---|---|---|
| | UT | Code | UT | Code | UT | Code | UT | Code | UT | Code |
| CURE-14B | 0.276 (0.008) | 0.088 (0.0042) | 0.125 (0.010) | 0.018 (0.0025) | 0.274 (0.012) | 0.064 (0.0029) | 0.206 (0.015) | 0.046 (0.0031) | 0.328 (0.038) | 0.029 (0.0013) |
| CURE-7B | 0.177 (0.005) | 0.062 (0.0037) | 0.387 (0.030) | 0.031 (0.0021) | 0.201 (0.009) | 0.047 (0.0032) | 0.258 (0.014) | 0.045 (0.0036) | 0.205 (0.022) | 0.023 (0.0011) |
| CURE-4B | 0.478 (0.009) | 0.021 (0.0034) | 0.068 (0.010) | 0.011 (0.0021) | 0.359 (0.021) | 0.014 (0.0025) | 0.286 (0.004) | 0.023 (0.0027) | 0.117 (0.024) | 0.025 (0.0014) |

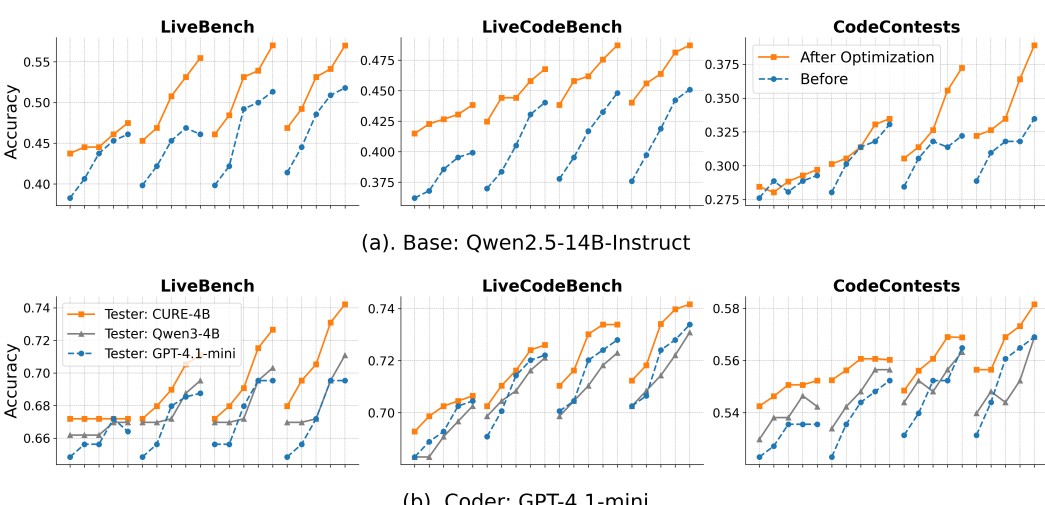

(a). Base: Qwen2.5-14B-Instruct

(b). Coder: GPT-4.1-mini

Figure 6: BoN performance improvement across benchmarks. Four curves (left to right) show sampling 2, 4, 8, and 16 generated codes; each curve's five points represent 1, 2, 4, 8, and 16 generated unit tests. (a). Improvement in BoN performance on open-source models after optimization. (b). BoN improvement with optimized unit tester on GPT-series coders.

Table 4: Response length (in tokens) of Qwen3-4B and CURE-4B in unit test generation task, corresponding to Figure 1 (e).

| Benchmark | Qwen3-4B | CURE-4B |
|---|---|---|
| LiveBench | 4711 | 3067 |
| MBPP | 2419 | 1611 |
| LiveCodeBench | 4326 | 2837 |
| CodeContests | 6086 | 3899 |
| CodeForces | 7309 | 4706 |

# C  Details of Experiments

## C.1  Detailed Algorithm

We have our detailed CURE optimization pipeline as follows. The training data we use is our training split of CodeContests. We set $n = m = 32$, $\eta = 1e-6$ and $\beta = 0.01$.

**Algorithm 1 CURE**

1: **Input:**
2:     1) A set of coding tasks $D = \{q_1, q_2, \ldots, q_N\}$.
3:     2) A poliy $\pi_\theta$ parameterized by $\theta$.
4:     3) Number of iterations $M$.
5:     4) Number of code solutions generated in each step: $n$.
6:     5) Number of unit tests generated in each step: $m$.
7:     5) Learning rate $\eta$, KL coefficient $\beta$.
8: **Initialize:** Policy parameters $\theta$.
9: **for** $t = 1$ to $M$ or not converged **do**
10:     **Collect rollout samples:**
11:     **for** each task $q \in D$ **do**
12:         Generate $n$ code solutions, $s_j$, $1 \leq j \leq n$, by policy $\pi_\theta$.
13:         Generate $m$ unit tests, $u_k$, $1 \leq k \leq m$, by policy $\pi_\theta$.
14:         Executing the $n$ generated solutions against these $m$ unit tests produces a binary evaluation matrix $\mathcal{B} \in \{0,1\}^{n \times m}$.
15:     **end for**
16:     **Obtain the reward for code solutions:**
17:     **for** each task $s_j$, $1 \leq j \leq n$ **do**
18:

$$\mathcal{R}^\star_{s_j} = \sum_{l=1}^{t_q} \mathcal{B}^\star_{j,m+l}$$

19:     **end for**
20:     **Obtain the reward for each unit test:**
21:     **for** each task $u_k$, $1 \leq k \leq m$ **do**
22:

$$\mathcal{R}^\star_{u_k} = -\sum_{l=1}^{n}(1 - \mathcal{I}_{s_l})\mathcal{B}^\star_{l,k} + (\prod_{l=1}^{n}\mathcal{I}_{s_l}\mathcal{B}^\star_{l,k})(\sum_{l=1}^{n}(1 - \mathcal{I}_{s_l}))$$

23:         **if** $\pi_\theta$ is long-cot model **then**
24:             $\mathcal{R}^\star_{u_k} = trans(\mathcal{R}^\star_{u_k}, l_k)$
25:         **end if**
26:     **end for**
27:     **Optimize the policy $\pi_\theta$:**

$$\mathcal{J}(\theta, \{o_i\}_{i=1}^{G}) = \mathbb{E}_{\substack{q \sim P(Q) \\ \{o_i\}_{i=1}^{G} \sim \pi_{\theta_{\text{old}}}(\cdot|q)}}\left[\frac{1}{G}\sum_{i=1}^{G}\min\left[\frac{\pi_\theta(o_i \mid q)}{\pi_{\theta_{\text{old}}}(o_i \mid q)}A_{o_i}, \ \text{clip}\left(\frac{\pi_\theta(o_i \mid q)}{\pi_{\theta_{\text{old}}}(o_i \mid q)}, \varepsilon\right)A_{o_i}\right]\right]$$

$$-\mathbb{E}_{\substack{q \sim P(Q) \\ \{o_i\}_{i=1}^{G} \sim \pi_{\theta_{\text{old}}}(\cdot|q)}}\left[\beta\, \text{D}_{\text{KL}}\big[\pi_\theta \,\|\, \pi_{\text{ref}}\big]\right],$$

28:     Fine-tune $\pi_\theta$ to obtain updated parameters $\theta \leftarrow \theta - \eta\nabla_\theta\mathcal{J}(\theta, \{s_j\}_{j=1}^{n})$,
29:     where $A_{s_j} = normalize(\mathcal{R}^\star_{s_j})$.
30:     Fine-tune $\pi_\theta$ to obtain updated parameters $\theta \leftarrow \theta - \eta\nabla_\theta\mathcal{J}(\theta, \{u_k\}_{k=1}^{m})$,
31:     where $A_{u_k} = normalize(\mathcal{R}^\star_{u_k})$.
32: **end for**
33: **Output:** Trained generator $\pi_\theta$.

## C.2 Prompt Design

This is the prompt for code generation:

This is the prompt for unit test generation:

## C.3  Preprocess Data

In our experiments, we adopt the stdio format for inputs and outputs, which is the standard input/output format used in LiveBench [45], LiveCodeBench [19], CodeContests [23], and CodeForces [31]. However, some tasks in LiveBench and LiveCodeBench, as well as all tasks in MBPP [2], originally use a functional input/output format. For consistency and ease of evaluation, we convert these functional formats to stdio. Specifically, the conversion rule is as follows: each variable is placed on a separate line, and lists are flattened into space-separated values on a single line, as illustrated in the following example:

**Input and output format example**

```
# functional format:

assert work("a", [1, 2, 3]) == 2

# stdio format:

Input:

a

1 2 3

Output:

2
```

For evaluation, we directly use the ground-truth code provided in CodeContests and MBPP. For Codeforces, LiveCode, and LiveCodeBench, we collect code generated by QwQ-32B [40] (using BoN with a maximum of 3 samples) that passes all ground-truth tests to serve as the ground-truth code.

## C.4   Test-time Scaling and Agentic Coding

We introduce how we apply MPSC [16], AlphaCodium [35] and S* [21] in our test-time scaling and agentic coding applications.

**MPSC**   For each task, we generate 8 samples of code, unit tests, and specifications (A specification is a pair of functions—a pre-condition and a post-condition—that define the valid input space and the expected input-output behavior of a program, serving as a formal description of its intended functionality.). We then follow the iterative optimization algorithm to derive the consistency scores, which will be used to identify the optimal code solution.

**AlphaCodium**   Following their procedure, we generate 8 code solutions per task using reasoning over public tests, along with 8 corresponding unit tests. Each code solution undergoes 2 iterations of refinement based on execution results from the public tests, followed by another 2 iterations based on execution results using the generated unit tests. Specifically, the refinement step asks the model to check the unit tests, code, and execution results, and then decide whether to refine or not.

**S***   We generate 8 code solutions and apply 4 iterations of self-debugging using public tests to obtain 8 refined versions. Note that the debugging is based on the execution results of ground-truth unit tests, so we directly ask the model to modify the code if the execution fails. The final solution is selected via their pairwise comparison method, using generated unit tests for evaluation.

## C.5   Agentic Unit Test Generation Methods

**We first introduce the development of unit test generation methods.**   Traditional approaches rely on software analysis techniques such as search-based methods (Evosuite) [11], random testing (Randoop) [30], model checking [9, 12], and symbolic execution [33, 10]. To improve scalability, neural machine translation-based methods were introduced [41, 1]. Specifically, AthenaTest [41] employs a BART model, while A3Test [1] uses a PLBART model with post-processing for improved accuracy. With the recent advancements in LLMs, prompt-based agentic methods such as ChatTester [50], ChatUniTest [6], and TestART [13] have demonstrated superior performance, further highlighting the potential of training LLMs for unit test generation. In this paper, we adopt the iterative refinement and generation pipeline used in ChatTester and ChatUniTest.

**Detailed Approach for Agentic Unit Test Generation in Our Experiments**   For the function-specific unit test generation task, where the input includes both the task description and ground-truth code, we prompt the policy model to generate both the code and the corresponding unit test. We then execute them to obtain the test result. If the test passes, the process proceeds to the next iteration; if it fails, we provide the code, unit test, and execution feedback to the policy model, which decides whether to refine the unit test. The maximum number of iterations is set to 3.

## C.6   Generated Unit Test Examples

**Examples for CURE-14B**

> ### Task 1
>
> You are given an array of integers `nums` of length `n`.
> The **cost** of an array is the value of its first element. For example, the cost of `[1,2,3]` is 1, while the cost of `[3,4,1]` is 3.
> You need to divide `nums` into 3 disjoint contiguous subarrays.
> Return the **minimum possible sum** of the cost of these subarrays.
> **Example 1:**
> **Input:**

```
1 2 3 12
```
**Output:**

6

**Explanation:** The best possible way to form 3 subarrays is: `[1]`, `[2]`, and `[3,12]` at a total cost of `1 + 2 + 3 = 6`.
Other possible ways include:

- `[1]`, `[2,3]`, `[12]` with a cost of `1 + 2 + 12 = 15`
- `[1,2]`, `[3]`, `[12]` with a cost of `1 + 3 + 12 = 16`

---

---

## Task 2

You are given a 0-indexed integer array `batteryPercentages` having length n, denoting the battery percentages of n 0-indexed devices.
Your task is to test each device `i` in order from 0 to `n - 1`, by performing the following test operations:

- If `batteryPercentages[i]` is greater than 0:
  - Increment the count of tested devices.
  - Decrease the battery percentage of all devices with indices `j` in the range `[i + 1, n - 1]` by 1, ensuring their battery never goes below 0, i.e., `batteryPercentages[j] = max(0, batteryPercentages[j] - 1)`.
  - Move to the next device.
- Otherwise, move to the next device without performing any test.

Return an integer denoting the number of devices that will be tested after performing the test operations in order.

**Example 1:**
**Input:**

1 1 2 1 3

**Output:**

3

**Explanation:**

- At device 0: batteryPercentages[0] > 0, so count = 1. batteryPercentages becomes [1,0,1,0,2]
- At device 1: batteryPercentages[1] == 0, skip.
- At device 2: batteryPercentages[2] > 0, count = 2. batteryPercentages becomes [1,0,1,0,1]
- At device 3: batteryPercentages[3] == 0, skip.
- At device 4: batteryPercentages[4] > 0, count = 3.

So the answer is 3.

---

Generated unit test for task 2

1. Start with a case where devices alternate between having battery and not having battery to ensure we understand the decrement logic correctly.
2. Let's consider the input [1, 0, 2, 0, 3].

**Analysis:**

- At device 0: batteryPercentages[0] = 1 > 0
  - Increment tested count: 1
  - Decrement indices 1 to 4 → batteryPercentages becomes [1,0,1,0,2]
- At device 1: batteryPercentages[1] = 0 → skip
- At device 2: batteryPercentages[2] = 1 > 0
  - Increment tested count: 2
  - Decrement indices 3 to 4 → batteryPercentages becomes [1,0,1,0,1]
- At device 3: batteryPercentages[3] = 0 → skip
- At device 4: batteryPercentages[4] = 1 > 0 → increment tested count: 3

**Test Input:**

1 0 2 0 3

**Test Output:**

3

**Explanation:**

- Step-by-step execution results in final batteryPercentages: [1,0,1,0,1]
- Devices tested: 0, 2, and 4 → total = 3

---

Task 3

You are given an array `nums` consisting of positive integers.
Return the total frequencies of elements in `nums` such that those elements all have the maximum frequency.
The frequency of an element is the number of occurrences of that element in the array.
**Example 1:**
**Input:**

1 2 2 3 1 4

**Output:**

4

**Explanation:** The elements 1 and 2 have a frequency of 2, which is the maximum frequency in the array.
So the number of elements in the array with maximum frequency is 2 (from 1) + 2 (from 2) = 4.

---

## Generated unit test for task 3

Let's consider an input where we have multiple elements with different frequencies. For instance, an array with several 2s and 3s, and a few 1s and 4s.
**Test Input:**

1 2 2 3 3 3 4 4 4 4

**Step-by-Step Computation:**

- Count the frequency of each element:
    - 1: 1 time
    - 2: 2 times
    - 3: 3 times
    - 4: 4 times
- Maximum frequency = 4 (from element 4)
- Only element 4 has this maximum frequency.

**Test Output:**

4

**Explanation:** The element 4 appears 4 times, and this is the highest frequency in the array. Therefore, the total number of elements with the maximum frequency is 4.

---

## Task 4

You are given an array of integers `nums` of length `n`.
The **cost** of an array is the value of its first element. For example, the cost of `[1,2,3]` is 1, while the cost of `[3,4,1]` is 3.
You need to divide `nums` into 3 disjoint contiguous subarrays.
Return the **minimum possible sum** of the cost of these subarrays.
**Example 1:**
**Input:**

1 2 3 12

**Output:**

6

**Explanation:**

- The best way to partition is `[1]`, `[2]`, `[3,12]` with a total cost: 1 + 2 + 3 = 6.
- Other possibilities:
    - `[1]`, `[2,3]`, `[12]` → cost = 1 + 2 + 12 = 15
    - `[1,2]`, `[3]`, `[12]` → cost = 1 + 3 + 12 = 16

---

## Generated unit test for task 4

**1. Designing the Input:**

- Use an array of length 5 to explore multiple partitioning scenarios.

- Mix small and large integers for diversity.

Consider the array [4, 1, 5, 2, 6].

**2. Computing the Output:** Try all valid 3-way partitions:

- [4], [1], [5, 2, 6] → cost = 4 + 1 + 5 = 10
- [4], [1, 5], [2, 6] → cost = 4 + 1 + 2 = 7
- [4], [1, 5, 2], [6] → cost = 4 + 1 + 6 = 11
- [4, 1], [5], [2, 6] → cost = 4 + 5 + 2 = 11
- [4, 1], [5, 2], [6] → cost = 4 + 5 + 6 = 15
- [4, 1, 5], [2], [6] → cost = 4 + 2 + 6 = 12

Minimum cost = 7 from partition [4], [1, 5], [2, 6].

**Test Input:**

4 1 5 2 6

**Test Output:**

7

**Explanation:** The optimal partition is:

- [4] → cost = 4
- [1, 5] → cost = 1
- [2, 6] → cost = 2

Total cost: 4 + 1 + 2 = 7.

---

**Examples for CURE-4B (long-CoT)**

Task 5

You are given an array of integers nums of length n.
The **cost** of an array is the value of its first element. For example, the cost of [1,2,3] is 1 while the cost of [3,4,1] is 3.
You need to divide nums into 3 disjoint contiguous subarrays.
Return the **minimum possible sum** of the cost of these subarrays.
**Example 1:**
**Input:**

1 2 3 12

**Output:**

6

**Explanation:** The best possible way to form 3 subarrays is: [1], [2], and [3,12] at a total cost of 1 + 2 + 3 = 6. The other possible ways to form 3 subarrays are:

- [1], [2,3], [12] → cost = 1 + 2 + 12 = 15
- [1,2], [3], [12] → cost = 1 + 3 + 12 = 16

Generated unit test for task 5

<think>
Okay, let's see. I need to create a new test case for this problem. The problem is about selecting up to L subjects without overlapping, maximizing happiness. So I need to make sure the input follows the constraints and that the output is correct.

First, I need to make sure the input is valid. Let's start by choosing small values for N, M, L that are within the constraints. Let's say N is 3, M is 3, L is 2. That's manageable.

Now, I need to create courses that don't overlap and can be selected without conflict. Let me think of some courses that are non-overlapping. For example:

Course 1: di=0 (Monday), ai=1, ki=1 (so runs during 1st period). That's 1 period. Then course 2 could be di=0, ai=2, ki=1. Then course 3 could be di=1, ai=1, ki=1. Then they can be selected as they don't overlap. But since L is 2, I need to select 2 courses that give maximum happiness.

Let me think of their ti values. Let's say course 1 has ti=5, course 2 has ti=4, course 3 has ti=3. Then the max would be 5+4=9. But maybe there's a better selection.

Alternatively, if there's a course that overlaps with others. Wait, but in this case, the courses are not overlapping. But maybe in the test case, it's possible to select two courses that don't conflict, but another combination gives higher happiness.

Alternatively, maybe there's a scenario where overlapping courses can't be selected. So I need to design courses that allow different combinations.

Another approach: Let me think of two courses that can be chosen. For instance, course 1 (d=0, a=1, k=1) and course 2 (d=1, a=1, k=1). But course 1 is on Monday, a=1, so runs from Monday period 1, and course 2 is Tuesday period 1. So they are not overlapping. They can be both selected if L is 2. So total happiness is sum of their ti.

But maybe another course has higher ti. Let me try to make a test case where the best selection is two courses that don't overlap, but perhaps have higher ti.

Let's design a scenario. For example, N=3 (days 0-4?), wait N is the number of periods per day. Wait the problem says that classes are from Monday to Friday (5 days) and for each day, there are N periods (from 1 to N), and courses start at ai and run ki frames (consecutive). So for example, if ai is 1 and ki is 2, the course runs from 1 to 2 (periods 1 and 2) of that day.

So for the test case, I need to make sure that courses are not overlapping. For example, a course that starts on Monday (d=0) at a=1, ki=1 (so runs Monday period 1), and a course that starts on Tuesday (d=1) at a=2, ki=2 (so runs Tuesday period 2 and 3) — they don't overlap. But to make this work, let's create a few courses that can be selected and have different ti values.

Let me try to create a test case:
N=3 (so each day has 3 periods)
M=3, L=2.
Course1: d=0 (Monday), a=1, k=1, t=5. It runs from Monday period 1.
Course2: d=0 (Monday), a=2, k=1, t=6. It runs from Monday period 2.
Course3: d=1 (Tuesday), a=1, k=1, t=4. It runs on Tuesday period 1.
Now, if L is 2, the best selection is course1 and course3 (t=5+4=9) or course2 and course3 (6+4=10). But course1 and course2 overlap on Monday, so they cannot be selected together. So the maximum is course2 and course3, which sum to 6+4=10.

But wait, if course2 and course3 are selected, then course2 is on Monday period 2, course3 is on Tuesday period 1. They don't conflict, so that's allowed.

So in that case, the output would be 10.

But maybe that's the test case. However, I need to make sure that the input is correct.

Let me structure the input as:
N=3, M=3, L=2
Then three courses:
d1=0, a1=1, k1=1, t1=5
d2=0, a2=2, k2=1, t2=6
d3=1, a3=1, k3=1, t3=4
So the input is:
3 3 2
0 1 1 5
0 2 1 6
1 1 1 4
The output would be 6+4=10?
Yes, that would be the maximum. Because course2 and course3 can be taken.
But what if another combination? For example, course2 (t=6) and course3 (t=4) is 10.
Alternatively, course2 and course3 is allowed. Then that's the correct answer.
Now, the test input would be:
Test Input:
3 3 2

0 1 1 5
0 2 1 6
1 1 1 4

Then the output is 10. So that's a valid test case.

But I need to check whether the courses don't overlap. Course1 is Monday 1, course2 is Monday 2. So they don't conflict. However, course2 and course1 are on the same day, but different periods, so they can be taken together if L is larger than 1. But in this case, L is 2. But wait, the problem says that each student can take up to L subjects that do not overlap.

But in the above example, course1 and course2 are on the same day, same day, but different periods, but since the course starts from ai and runs for ki frames, they are consecutive. So course1 is Monday, period 1. It's for 1 frame. So the course is from Monday period 1. Course2 is Monday period 2, so it runs for 1 frame. So they are on the same day but not overlapping. So they can be selected as two different courses. But in that case, if the L=2, then the maximum would be 5+6=11? Wait, that's a mistake in the previous analysis. Oh, right! Because in this case, course1 and course2 can both be selected if they are on the same day, and not overlapping. So the maximum would be 5+6=11.

But then that would be a better selection. So the test case needs to have courses that are non-overlapping. So why would the answer be 10? That suggests that I made a mistake in the test case.

Wait, this is a problem. Let me think again.

So if the courses are on the same day, but their periods are non-overlapping, then they can be taken together. For example, course1 is on Monday, period 1, runs for 1 frame. Course2 is on Monday, period 2, runs for 1 frame. So they can be selected together. So the maximum is course1 + course2 (t=5+6=11) instead of course2 and course3 (t=6+4=10). So this is a problem.

Ah, I think my earlier test case is flawed because the maximum would be 5+6 = 11, which is higher than the previous thought. That suggests I need to adjust the test case to avoid such cases.

So, what's the correct way to create a test case where the maximum is not selecting overlapping courses on the same day.

Let me re-construct the test case. Let's have courses that cannot be selected together, even if they are on the same day.

For example, course1 is on Monday (d=0), ai=1, ki=2. That would cover periods 1 and 2 on Monday. Then course2 is on Monday (d=0), ai=2, ki=1. So the course1 starts on ai=1, runs for 2 frames. So it's periods 1 and 2. So course2 starts on period 2 and runs for 1 frame. That would overlap with course1, so they can't be taken together. So the user can't take course1 and course2 together.

So for that scenario, if L is 2, the maximum would be course1 (t=5) + course3 (t=4) = 9.

Let me try this.

Let's have:

N=3 (so periods are 1-3 each day)

M=3 courses:

Course1: d=0 (Monday), a=1, k=2, t=5. Runs on Monday 1 and 2.

Course2: d=0 (Monday), a=2, k=1, t=6. So starts on Monday 2. It overlaps with course1. So cannot be selected with course1.

Course3: d=1 (Tuesday), a=1, k=1, t=4. No overlap with course1. So if L is 2, then the best would be course1 (5) + course3 (4) = 9. Or course2 (6) + course3 (4) = 10, but course2 is on Monday 2, and course3 is on Tuesday. So that's allowed. But course1 and course2 can't be selected together. So the maximum is 6+4=10. Or course2 and course3.

But that's better than course1 and course3.

So in that case, the correct answer would be 10.

Now, this scenario makes sense. So the input would be:

N=3, M=3, L=2.

Then three courses:

0 1 2 5
0 2 1 6
1 1 1 4

Now, the courses are as follows:

course1: d=0 (Monday), starts period 1, runs for 2 frames (periods 1 and 2). So course1 covers Monday periods 1 and 2.

course2: d=0, a=2, k=1: period 2 of Monday, runs 1 frame. So this overlaps with course1.

course3: d=1 (Tuesday), period 1, runs 1 frame.

So, possible selections:

- course2 and course3: t=6+4=10.

- course3 and course1: 5+4=9.

- course2 and course3 is better.

So the correct answer would be 10.

In this case, the test case would be:

Test Input:

3 3 2

0 1 2 5

0 2 1 6

1 1 1 4

Then the Output is 10.

So this should be a valid test case, and the output is 10. That test case would allow the code to correctly distinguish between overlapping and non-overlapping courses.

I need to verify that each course's ai and ki are valid. For example, for course1, ai is 1, ki is 2. Then ai+ki-1 is 1+2-1=2. Which is <=3 (since N=3). So yes, valid.

For course2, ai=2, ki=1: ai + ki -1 = 2+1-1=2, which is within N=3.

So that's correct.

So this test case should be valid and the output is 10.

Therefore, the final test case is: </think>

**Test Input:**

3 3 2

0 1 2 5

0 2 1 6

1 1 1 4

**Test Output:**

10

**Explanation:** We have three courses. The first course runs for two periods (Monday, periods 1 and 2) and provides a happiness of 5. The second course runs for one period (Monday, period 2) and provides a happiness of 6, but overlaps with the first course. The third course runs for one period (Tuesday, period 1) and provides 4. The best combination is the second and third courses, which are non-overlapping. Their total happiness is 6 + 4 = 10, which is the maximum possible.

