# OpenReview forum: "Co-Evolving LLM Coder and Unit Tester via Reinforcement Learning"
_NeurIPS.cc/2025/Conference — NeurIPS 2025 spotlight_

### Official Review · Reviewer_q1zj · 2025-06-24

**Clarity:** 4
**Significance:** 3
**Originality:** 2
**Rating:** 5
**Confidence:** 4

**Summary:**

CURE is a framework for post-training and test-time scaling improvements in language models for code generation. The key idea is to have language models generate programs and tests simultaneously, optimizing the quality of both artifacts using a reinforcement learning approach and a clever reward function. The authors perform a thorough evaluation of CURE, finding that it generates better code and tests and complements existing approaches well.

**Questions:**

- What kind of programs and tests can this approach handle? Can it be extended to stateful programs, for example? What about flaky tests? What is the scope of this work, and can it eventually be extended beyond that?

**Ethical Concerns:**

["NO or VERY MINOR ethics concerns only"]

**Final Justification:**

As stated in my original review, the paper is solid and should be accepted.

The two main issues were that the scope of the work is not properly discussed and that the need for ground-truth unit tests is obscured. The authors provided an adequate response in the rebuttal and have agreed to include a discussion of their limitations in the final version. This will improve the paper, which I still believe should be accepted.

**Limitations:**

No. See question 1.

**Quality:**

3

**Strengths And Weaknesses:**

## Strengths
- The approach is sensible and well-motivated. The reward (4) may even be deeper than the authors let on. In the software engineering literature, there is a notion of “mutation testing” (see, e.g., https://ieeexplore.ieee.org/document/5487526) that is used to evaluate the quality of tests. The reward function is related to mutation testing but with the addition of agreement with ground truth tests. Very cool. Mutation testing has been used for LLM code generation before (see, e.g., the “completeness measure” of https://arxiv.org/pdf/2406.09757 that is very closely related to one half of the reward function), and the authors may enjoy exploring that space.
- The empirical evaluation is solid.
- The related work is honest, and the comparisons are apt.

## Weaknesses
- “without access to ground-truth code solutions” and “without any ground-truth code as supervision” are technically true but slightly misleading. The approach does require ground truth tests.
- The distinctions with related work are subtle. One might argue that the approach is slightly incremental. There is, of course, plenty of work on using LMs to generate code, tests, and combinations thereof.
The authors do not outline the scope of their work. See Question 1.

---

> ### Author Rebuttal · Authors · 2025-07-29
>
> *Thank you for your appreciation of this work! Please see below for our responses to your comments.*
>
>
> **Q1. “without access to ground-truth code solutions” and “without any ground-truth code as supervision” are technically true but slightly misleading. The approach does require ground truth tests.**
>
> Thanks for your constructive suggestion! We will highlight that we need ground-truth unit tests in abstract to aviod any misleading.
>
>
> **Q2. What kind of programs and tests can this approach handle? Can it be extended to stateful programs, for example? What about flaky tests? What is the scope of this work, and can it eventually be extended beyond that?**
>
> Our proposed CURE framework currently focuses primarily on deterministic coding tasks characterized by well-defined input-output behavior, such as problems typically evaluated with stdin/stdout-based UTs. Specifically, it targets generating and validating functional correctness through automatically generated UT derived directly from task descriptions. The general goal is to unify strong coding and unit test generation ability within a single agent efficiently to help develop robust and reliable coding systems capable of automated self-checking, self-correction in real-world programming tasks. We will add this scope in our discussion part.
>
> Regarding stateful programs, the current pipeline is inherently stateless. However, if the environment states can be explicitly managed or simulated during training, it can be adapted to handle stateful settings, enabling more real-world applications.
>
> Since our reward signals assume deterministic outcomes, handling flaky tests—which produce inconsistent results due to nondeterminism or environmental instability—would require extending our method with additional techniques to derive reliable reward signals, such as repeated test executions and methods to mitigate nondeterministic behaviors.
>
> **Q3. The distinctions with related work are subtle. One might argue that the approach is slightly incremental. There is, of course, plenty of work on using LMs to generate code, tests, and combinations thereof. The authors do not outline the scope of their work.**
>
> This work mainly focus on developing a efficient pipeline to jointly optimizing coder and unit tester within one unified model, without ground-truth code solution needed in training data, benefits plenty of downstream coding related tasks, as demonstrated in our experiments. The related work mainly focus solely on either agentic unit test generation, agentic coding, RL for coding task, or RL for unit test generation. Our pipeline provide a bridge to these tasks.

---

> > ### Comment · Reviewer_q1zj · 2025-08-05
> >
> > As stated in my original review, the paper is solid and should be accepted.
> >
> > The two main issues were that the scope of the work is not properly discussed and that the need for ground-truth unit tests is obscured. The authors provided an adequate response in the rebuttal and have agreed to include a discussion of their limitations in the final version. This will improve the paper, which I still believe should be accepted.

---

> > > ### Author Response · Authors · 2025-08-05
> > >
> > > Dear Reviewer q1zj,
> > >
> > > Thank you for your positive feedback. We will incorporate these analyses from the rebuttal to strengthen our paper. Your suggestion has been invaluable in improving our paper.
> > >
> > > Warm Regards,
> > >
> > > The Authors

---

### Official Review · Reviewer_VqpS · 2025-06-30

**Clarity:** 4
**Significance:** 3
**Originality:** 3
**Rating:** 4
**Confidence:** 4

**Summary:**

This paper introduces CURE, a novel co-evolutionary reinforcement learning framework that enhances code generation and unit test generation capabilities in large language models (LLMs). The main contribution of this work is its ability to co-evolve code generation and unit test generation without relying on ground-truth code solutions, which significantly improves scalability and flexibility. The proposed model is evaluated on five benchmarks, demonstrating a 6.2% improvement in code generation accuracy, a 25.1% improvement in test generation accuracy, and a 35% reduction in inference time for long-chain reasoning models.

**Questions:**

1. Do you agree with the theoretical flaw in Proposition A.1? If not, please provide a detailed explanation. The theoretical issue is fundamental to the reward system, and we will reassess the score based on your input.

2. Does the proposed reward function adequately differentiate between high-quality and low-quality unit tests? Could it inadvertently lead to the generation of less useful tests, such as overly permissive ones that fail to provide sufficient differentiation?

3. What are the experimental results under the correct reward?

If the authors can provide more details to address my comments, I would be inclined to raise my score.

**Ethical Concerns:**

["NO or VERY MINOR ethics concerns only"]

**Final Justification:**

Based on the provided `reward.py` code and the authors' rebuttal, we confirm that the reward logic is consistent with the intended design.

- Unit tests are evaluated based on whether they pass all *correct* code samples (those that pass all ground-truth unit tests), not all generated code. This matches the explanation given in the rebuttal.

- The issue in Proposition A.1 appears to be a typographical error in the text, not a flaw in the actual implementation.

- The implementation includes mechanisms to reward unit tests that can detect flawed code, encouraging more discriminative and useful test cases.

- The code supports efficiency-aware training via length normalization when `enable_efficient` is enabled, as described by the authors.

We believe the mismatch was indeed a writing error. The code reflects the correct logic, and the explanation is acceptable. Overall, the implementation quality meets the expected standard.

**Limitations:**

Yes

**Paper Formatting Concerns:**

There are no major formatting issues observed in this paper. The paper is well-formatted and adheres to the required NeurIPS standards.

**Quality:**

3

**Strengths And Weaknesses:**

### Strengths
- **Originality**: The idea of using co-evolutionary reinforcement learning to simultaneously optimize code generation and unit test generation is novel and addresses the significant challenge of training models without ground-truth code.
- **Impact**: The model demonstrates considerable improvements in both code generation and unit test generation across multiple benchmarks, which is highly relevant for practical software development and automation.
- **Scalability**: The removal of the reliance on labeled data opens the potential for scaling this approach to a wider range of real-world applications.

### Weaknesses
- **Theoretical Flaws**: Proposition A.1 contains a critical error. It incorrectly assumes that the quality of a unit test can be directly tied to the quality of the code. This leads to an issue where the evaluation of a test quality is overly dependent on the code's accuracy, ignoring that a perfect test might still be penalized if the code fails in certain ways.

  **Example**: A perfect unit test (with $p_u$ = 1) could be incorrectly evaluated as  $\hat{p_u}$ = 0  if the code implementation is flawed, even though the test itself is correct. This misjudgment undermines the reliability of the reward system.

- **Reward Function Issues**: The reward function may penalize high-quality unit tests, like boundary cases, while rewarding simpler, less effective tests. This could lead to suboptimal performance.
- **Experimental Reliability**: The reward system might lead to flawed test generation. Additional experiments are needed to ensure the model can avoid producing degenerate tests (e.g., tests that pass all code but lack discriminative power).

---

> ### Author Rebuttal · Authors · 2025-07-29
>
> *Thanks for your careful review and detailed comments! Please see below for our responses to your comments.*
>
> **Q1. Proposition A.1 assumes quality of unit test can be directly tied to quality of code. A perfect unit test ($p\_u = 1$) could be incorrectly evaluated if the code is flawed ($\hat{p}\_u = 1$).**
>
> Sorry for the confusion caused by this typo! The formula we used is
> $\hat{p}\_u = \prod\_{l: I\_{s\_l} = 1} I\_{s\_l} B\_{l, k}^{\star}$,  while we missed the index in writing:
> $\hat{p}\_u = \prod\_{l = 1}^n I\_{s_l} B\_{l, k}^{\star}$, which led to the misunderstanding. The reward system we use here is sound and correct: a unit test is judged as correct (positive) if and only if it passes all correct codes that have been verified by all ground-truth unit tests (provided in the dataset, we also conducted quality control to ensure correctness). Meanwhile, the scale of the reward is determined by the execution results on the incorrect (flawed) codes.
>
> To clarify this was indeed a typo:
>
> (1). **As stated in lines 172–181 of the paper**: When UT passes **all correct** codes ($\hat{p}\_u = 1$), the reward for UT (unit test) is positive and proportional to number of flawed codes it can detect. When UT fails on **any correct** code ($\hat{p}\_u = 0$), the reward for UT is negative and proportional to number of flawed codes it passes. Here, correctness of code is defined by passing all ground-truth unit tests. And the correctness of generated unit test is defined by passing all correct codes: $\hat{p}\_u = \prod\_{l: I\_{s\_l} = 1} I\_{s\_l} B\_{l, k}^{\star}$, not $\hat{p}\_u = \prod\_{l = 1}^n I\_{s\_l} B\_{l, k}^{\star}$.
>
> (2). **In the explicit reward assignment example in Figure 2** of the paper, the reward is calculated exactly by the correct version. If the typo version were used, there would be no positive reward. Because in this example, there are 3 flawed codes among 6 generated codes. In the typo version, the rewards for the first three unit tests would be 0, -1, and -2, instead of 3, 2, and 1 as shown in the figure.
>
> (3). **You can also verify this** by checking our submitted code at `CURE/optimization/reward.py`.
>
> (4). Finally, we train Qwen2.5-7B-Instruct using the typo-version reward and obtained poor results (Table 1 here), which are completely different from those in paper.
>
> We sincerely thank the reviewer for noticing this typo. We will fix the typo and replace it with the correct $\hat{p}\_u$ in our unit test reward $R\_{u\_k}^{\star}$. Our reward design ensures that the correctness of a UT is only estimated by the execution results on ground-truth code, not on all generated code.
>
> **Table 1. UT acc before and after training**
> ||LiveBench|CodeContests|
> |-|-|-|
> |before|26.5|26.7|
> |our reward|44.2|52.5|
> |typo reward|24.8|26.0|
>
>
> **Q2. Does the reward adequately differentiate between high and low-quality UTs? Could it lead to generation of less useful tests, such as permissive ones that provide less differentiation? Reward may penalize high-quality unit tests. Additional experiments are needed to ensure the model can avoid producing degenerate tests (pass all code but lack discriminative power).**
>
> (1). From a theoretical view, our reward considers this situation by letting the reward for UT as $-\sum\_{l = 1}^n (1 - I\_{s\_l}) B\_{l, k}^{\star} + \hat{p}\_{u\_k} \sum\_{l = 1}^n (1 - I\_{s\_l})$, which is proportional to the number of flawed codes it can detect when the UT is detected as correct ($\hat{p}\_{u\_k} = 1$), specifically calculated as $\sum\_{l = 1}^n (1 - I\_{s\_l}) (1 - B\_{l, k}^{\star})$. This design ensures that a more discriminative correct UT receives a higher reward than those that are correct but less discriminative. The reward does not penalize these high-quality unit tests; instead, it rewards them more.
>
> (2). As an additional experiment for demonstration, for each task in the testing set, we sample 8 correct unit tests (verified by ground-truth codes; if 8 are not reached, we keep generating) using both Qwen2.5-7B-Instruct and Qwen2.5-7B-Instruct after optimization. For these two groups of unit tests, we use them to perform the BoN(8) task for Qwen2.5-7B-Instruct as a common coder and obtain the results shown in Table 2. The BoN strategy exactly follows the approach described in the paper, selecting the single best code among eight independently generated codes that passes the greatest number of the eight generated unit tests. We find that the trained tester can better select the codes to achieve higher BoN acc, even when all UTs used here are correct. This rigorously demonstrates that the reward teaches the model to generate higher-quality tests. We will add this experiment in our paper.
>
> **Table 2. BoN(8) acc**
> |tester|LiveCodeBench|CodeContests|LiveBench|
> |-|-|-|-|
> |before training|36.3|27.1|37.8|
> |after training|40.5|28.6|39.2|
>
>
> **Q3. What are the experimental results under the correct reward?**
>
> The reward used in our experiments is the correct reward rather than the typo-version reward, as demonstrated in the first response. We also demonstrated that our reward will not penalize the high-quality UT by experiments in response to Q2. We thank the reviewer for their careful attention and efforts to refine this work. If the reviewer is referring to "correct reward" as something different, please feel free to elaborate, and we will address your concerns with additional experiments.

---

> > ### Comment · Reviewer_VqpS · 2025-08-03
> >
> > Thank you for the clarification. After reviewing the `reward.py` implementation, I agree that the issue in Proposition A.1 was a typographical error rather than a flaw in the actual logic. The reward function is implemented as described in the rebuttal, and the overall quality of the implementation meets the expected standard.

---

> > > ### Author Response · Authors · 2025-08-03
> > >
> > > Dear Reviewer,
> > >
> > > Thank you for your positive feedback and we are pleased that our rebuttal has addressed your concerns. Your detailed reviews are valuable for improving our paper.
> > >
> > > Warm Regards,
> > >
> > > The Authors

---

### Official Review · Reviewer_SQc4 · 2025-07-03

**Clarity:** 3
**Significance:** 3
**Originality:** 2
**Rating:** 4
**Confidence:** 4

**Summary:**

This paper proposes a reinforcement learning framework that jointly improves an LLM coder and a unit-test generator without ground-truth code solutions. In every training round, the policy samples $n$ candidate programs and $m$ candidate tests for the same problem, forms an $n\times m$  execution matrix, and assigns:
- coder reward: the number of public (ground‑truth) tests passed;
- unit-test generator reward: a theory‑derived metric $\mu = p_{u} (1 - p_{01}) - (1 - p_{u}) p_{00}$ that favors unit tests which pass correct code solutions and reject wrong code solutions.

Experiments on five benchmarks demonstrate the effectiveness of the proposed approach. Moreover, the unit-test generator can serve as a reward model in further reinforcement learning loops.

**Questions:**

See Weaknesses.

**Ethical Concerns:**

["NO or VERY MINOR ethics concerns only"]

**Final Justification:**

This paper makes a solid contribution and should be accepted. The concerns raised during the review process have been properly addressed.

**Limitations:**

Yes

**Quality:**

4

**Strengths And Weaknesses:**

**Strengths:**

1.   Insightful theorem analysis. The paper gives a concise and rigorous derivation of the necessary and sufficient condition for the pairwise reward ordering to become almost surely correct
2.   Facilitates Label-free RL. The proposed approach leverages mutual supervision between code solutions and unit tests to improve both the code generator and unit-test generator, representing a significant step toward truly label-free RL.
3. Comprehensive evaluation. Results span five public benchmarks, multiple BoN settings, agentic pipelines, and API models, demonstrating consistent gains and meaningful inference‑time speed‑ups for long‑CoT testers.

**Weaknesses:**

1.   Discrepancy between Theorem 3.1 and the actual method. Theorem 3.1 implies that no ground-truth code solutions or unit tests are needed, yet in practice, the framework still heavily relies on ground-truth tests to estimate $p_u$, $p_{00}$, and $p_{01}$. Exploring approaches that eliminate the need for ground-truth tests (e.g., CodeT) would likely yield more convincing results.

2.   Weak co-evolution coupling.  Although the paper advertises a co-evolving coder and tester, in practice the coder’s reward still comes directly from ground-truth unit tests, and the tester’s reward assumes that any code passing those same tests is correct to estimate $p_u$, $p_{00}$, and $p_{01}$. This reliance on the same public test suite for both roles means the “co-evolution” is more superficial than synergistic.

---

> ### Author Rebuttal · Authors · 2025-07-29
>
> *Thank you for your appreciation of this work! Please see below for our responses to your comments.*
>
> **Q1. Discrepancy between Theorem 3.1 and the actual method. Theorem 3.1 implies that no ground-truth code solutions or unit tests are needed, yet in practice, the framework still heavily relies on ground-truth tests. Exploring approaches that eliminate the need for ground-truth tests would yield more convincing results.**
>
> Sorry for the misleading. The goal of Theorem 3.1 is to derive the objective we need to optimize in order to achieve BoN accuracy without relying on ground-truth unit tests (UT) during the inference stage. It does not imply that the optimization method does not require ground-truth UT too. As stated in Theorem 3.1, $\mu = p\_u (1 - p\_{01}) - (1 - p\_u) p\_{00} > 0$ is needed to ensure that the reward precision approaches 1. Only in this case is ground-truth UT not necessarily required to achieve high selection accuracy (can be replaced by plenty generated UTs). Theorem 3.1 also implies that $\mu$ determines the convergence rate of the reward precision, and a larger $\mu$ will make the optimization more efficient.
>
> For models not trained with our proposed method, $\mu > 0$ does not necessarily hold, or it may be too small, which means the selection accuracy is not guaranteed when only a limited number of UTs are sampled. This is why we need to use CURE with ground-truth UT to increase $\mu$.
>
> In conclusion, the results of Theorem 3.1 imply that we need ground-truth UT during optimization to increase the overall $\mu$ in order to achieve both reliable performance and efficiency, so that the optimized model has capability to do BoN inference without ground-truth UTs.
>
>
> However, we find that CURE is inspiring as a cold start, enabling the optimized model to co-evolve effectively without any ground-truth unit tests. We add two additional experiments here. After deriving Qwen2.5-7B-140, which refers to Qwen2.5-7B-Instruct after 140 steps of CURE optimization, we let it co-evolve 140 steps without any ground-truth UT by selecting “correct” codes in CodeT’s way. We find that this achieves similar improvements compared to continuing training with ground-truth UT. This indicates that as long as $\mu$ is large enough, the models can co-evolve without ground-truth UT, which aligns with Theorem 3.1.
>
>
> **Table 1. UT, code, and BoN(16) acc on LiveBench, after optimizing Qwen2.5-7B-140 with and without ground-truth UT**
> |Method|UT|Code|BoN|
> |-|-|-|-|
> |Before|44.2|37.3|45.5|
> |No ground-truth|54.5|37.9|51.2|
> |With ground-truth|54.8|38.2|51.4|
>
>
> **Q2. Weak co-evolution coupling. Although the paper advertises a co-evolving coder and tester, in practice the coder’s reward still comes directly from ground-truth unit tests, and the tester’s reward assumes that any code passing those same tests is correct to estimate.**
>
> We acknowledge the potential misleading nature of the term co-evolving, and we will emphasize in the abstract that ground-truth UTs are required. We understand that training from scratch without ground-truth UT may seem more inspiring and elegant, which was what we initially attempted. However, we found that the unoptimized model, Qwen2.5-7B-Instruct, has overly low UT accuracy and insufficient discriminative power to detect flawed codes, making co-evolving without ground-truth UT (using the same method as in Response 1) far less efficient than our pipeline, which relies on ground-truth UT (see Table 2). This observation aligns perfectly with the condition stated in Theorem 3.1.
>
> **Table 2. UT, code, and BoN(16) acc on LiveBench, after optimizing Qwen2.5-7B-Instruct with and without ground-truth UT**
> |Method|UT|Code|BoN|
> |-|-|-|-|
> |Before|26.5|31.1|35.9|
> |No ground-truth|35.8|33.5|39.2|
> |With ground-truth|44.2|37.3|45.5|

---

> > ### Comment · Reviewer_SQc4 · 2025-08-05
> >
> > I appreciate the authors for addressing my concerns; therefore, I would like to keep my positive score.

---

> > > ### Author Response · Authors · 2025-08-05
> > >
> > > Dear Reviewer SQc4,
> > >
> > > We are pleased that our rebuttal has addressed your concerns. Your feedback has been invaluable in improving our paper.
> > >
> > > Warm Regards,
> > >
> > > The Authors

---

### Official Review · Reviewer_1BiQ · 2025-07-03

**Clarity:** 4
**Significance:** 4
**Originality:** 4
**Rating:** 5
**Confidence:** 3

**Summary:**

The paper proposes CURE, which is an RL framework, to jointly optimize a single model for code generation and writing unit tests. The framework only depends on ground-truth test cases and does not require any code solutions. The authors evaluate their approach for a variety of models across 5 public datasets.

**Questions:**

- Can you comment on what happens during reward estimation if the ground-truth test-cases are either not sound or not complete? Specially in cases where the initial test-cases are minimal and incomplete, the tester seems to be penalized for catching buggy code.

- In Figure 5, is CURE originally trained on the same unit-tests that are also used in the ‘labeled RL’ baseline?

**Ethical Concerns:**

["NO or VERY MINOR ethics concerns only"]

**Final Justification:**

One important issue was making it more clear that the approach relies on high quality ground truth test cases. Authors have addressed this in the rebuttal. They also added more results on the nature of the generated test cases and noted the scope for future work. Overall, the approach is sound, and I would like to maintain my original accept rating.

**Limitations:**

Additional discussion on how the current evaluations are limited to algorithmic coding tasks in Python is needed.

**Paper Formatting Concerns:**

None.

**Quality:**

3

**Strengths And Weaknesses:**

Strengths:
- The considered setting where test-cases are more widely available over solutions is practically relevant for coding.
- The proposed reward is well motivated and properly justified.
- The paper provides rigorous evaluation, showing significant improvements across multiple datasets and tasks.

Weaknesses and Suggestions:
- Ablation on how many ground truth test-cases are needed and what happens when test-cases tend to zero is missing.
- Clarify in the abstract that the approach relies on ground-truth test-cases.
- Error bars are only available for a subset of the results.
- A qualitative analysis on the types of unit-tests generated by CURE would be useful to show that they go beyond trivial cases, and what they are still missing.

---

> ### Author Rebuttal · Authors · 2025-07-29
>
> *Thank you for your appreciation of this work! Please see below for our responses to your comments.*
>
>
> **Q1. Can you comment on what happens during reward estimation if the ground-truth test-cases are either not sound or not complete? Specially in cases where the initial test-cases are minimal and incomplete, the tester seems to be penalized for catching buggy code.**
>
> We conducted quality control on the CodeContests training data to avoid this issue by executing the ground-truth codes and ground-truth unit tests to ensure that all ground-truth unit tests are correct. We also filtered out tasks with fewer than 8 unit tests. If the ground-truth unit tests are still incorrect, the wrong code solution may be selected as the ground-truth code. If the ground-truth unit tests are still insufficient, some codes that only solve partial cases may receive positive rewards, which is a common challenge in coding RL settings. Both of these situations may affect the reward assigned to unit tests. One way to continually eliminate this problem is to keep generating more unit tests and confirm they are ground-truth through code execution, thereby augmenting the dataset further. We will add these discussions to the discussion section.
>
>
> **Q2. In Figure 5, is CURE originally trained on the same unit-tests that are also used in the ‘labeled RL’ baseline?**
>
> Yes. The reward model is also trained on the same datasets. If there is concern about whether this reward model can handle other training datasets, we add an additional experiment here. We compare labeled and unlabeled RL training on a different dataset, PrimeIntellect, for 100 epochs. Table 1 below shows the same conclusions.
>
> **Table 1. UT, code, and BoN(16) $\Delta$ acc on LiveBench for labeled and unlabeled RL**
> |Method| UT $\Delta$|Code $\Delta$|BoN $\Delta$|
> |-|-|-|-|
> |labeled|23.9|8.2|4.9|
> |unlabeled|20.7|7.1|4.2|
>
> **Q3. Ablation on how many ground truth test-cases are needed and what happens when test-cases tend to zero is missing.**
>
> We added the zero test-case setting as an additional experiment (see Table 2 in the response to Reviewer SQc4). We find that when ground-truth UT is missing, the training efficiency decreases. This observation aligns with Theorem 3.1, which states that a larger $\mu$ can improve the convergence rate of selection precision. Therefore, ground-truth UT is needed to improve training efficiency.
>
> We now filter training data with at least 16 unit tests, and train 100 steps with 8 unit tests and 16 unit tests to do the ablation study. From the following Table 2, we find that 8 unit tests are enough.
>
> **Table 2. Ablation studies, evaluated on LiveBench**
> |Method|UT|Code|BoN|
> |-|-|-|-|
> |8 UTs|42.4|37.0|44.6|
> |16 UTs|42.7|36.9|44.8|
>
>
>
> **Q4. A qualitative analysis on the types of unit-tests generated by CURE would be useful to show that they go beyond trivial cases, and what they are still missing.**
>
> We conduct an additional experiment to demonstrate our optimized model generates less trivial cases. For each task, we sample 8 correct unit tests (verified by ground-truth codes; if 8 are not reached, we keep generating) using both models, Qwen2.5-7B-Instruct, and Qwen2.5-7B-Instruct after optimization. For these two groups of unit tests, we use them to perform the BoN(8) task for Qwen2.5-7B-Instruct as a common coder and obtain the results shown in Table 3. We find that the trained tester can better select the codes to achieve higher BoN, even when all UTs used here are correct. This demonstrates that the optimizedd model tends to generate higher-quality tests. We also calculated the probability that these 8 codes contain at least one correct solution, denoted as *selection*, which is higher than "after training" setting. This indicates that there is still room for improving the discriminative power of the generated unit tests.
>
> **Table 3. BoN acc**
> |tester|LiveCodeBench|CodeContests|LiveBench|
> |-|-|-|-|
> |before training|36.3|27.1|37.8|
> |after training|40.5|28.6|39.2|
> |*selection*|42.6|30.3|42.1|
>
>
> **Q5. Error bars are only available for a subset of the results.**
>
>
> Due to computational limitations, we did not have time to calculate the error bars for the BoN settings, as this requires generating 16 codes and 16 UTs for each problem and executing 16 × 16 cases. Repeating this process to calculate error bars would be very challenging. We hope that the significant improvement achieved by our method can mitigate this limitation.
>
>
> **Q6. Clarify in the abstract that the approach relies on ground-truth test-cases.**
>
> Thanks for your constructive suggestion! We will hightlight that we need ground-truth unit tests in abstract to aviod any misleading.
>
>
>
> **Q7. Additional discussion on how the current evaluations are limited to algorithmic coding tasks in Python is needed.**
>
> We will add this discussion to the discussion section. Indeed, we only focus on Python tasks, like other coding RL projects. Implementing settings for other programming languages is an important research direction.

---

### Decision · Program_Chairs · 2025-09-17

**Decision:**

Accept (spotlight)

**Comment:**

The paper proposes CURE, a novel RL framework with a reward design that co-evolves code generation and unit test generation based on their interaction outcomes, without relying on ground-truth code as supervision. This enables flexible and scalable training, allowing unit testers to learn from coders and vice versa. Experiments show strong performance improvements for both code and unit test generation.

Strengths:
- Well-motivated reward design, grounded in software engineering literature (1BiQ, q1zj)
- Novel and scalable approach leveraging mutual supervision between code solutions and unit tests (SQc4, VqpS)
- Insightful theorem analysis for pairwise reward ordering (SQc4); a typo was noted but the authors already addressed in rebuttal (VqpS)
- Rigorous evaluation demonstrating significant improvements across multiple datasets and tasks (1BiQ, SQc4, VqpS, q1zj)

Weaknesses:
- Missing qualitative analysis of the types of unit tests generated, which would strengthen claims of non-trivial test generation (1BiQ)
- Co-evolution strategy is weaker than claimed; coder rewards still rely on ground-truth unit tests rather than being fully co-dependent (SQc4)

IN summary, I found CURE is a novel and well-justified framework that makes both conceptual and empirical contributions to code generation with RL. Despite some limitations around the strength of co-evolution, the work is strong overall and I recommended for acceptance.